# Learning Neural Contextual Bandits through Perturbed Rewards

**Yiling Jia[1], Weitong Zhang[2], Dongruo Zhou[2], Quanquan Gu[2], Hongning Wang[1]**
[1]Department of Computer Science, University of Virginia
[2]Department of Computer Science, University of California, Los Angeles.
`yj9xs@virginia.edu, wt.zhang@ucla.edu, drzhou@cs.ucla.edu,`
`qgu@cs.ucla.edu, hw5x@virginia.edu`

## Abstract

Thanks to the power of representation learning, neural contextual bandit algorithms demonstrate remarkable performance improvement against their classical counterparts. But because their exploration has to be performed in the entire neural network parameter space to obtain nearly optimal regret, the resulting computational cost is prohibitively high. We perturb the rewards when updating the neural network to eliminate the need of explicit exploration and the corresponding computational overhead. We prove that a $\widetilde{O}(\widetilde{d}\sqrt{T})$ regret upper bound is still achievable under standard regularity conditions, where $T$ is the number of rounds of interactions and $\widetilde{d}$ is the effective dimension of a neural tangent kernel matrix. Extensive comparisons with several benchmark contextual bandit algorithms, including two recent neural contextual bandit models, demonstrate the effectiveness and computational efficiency of our proposed neural bandit algorithm.

## 1 Introduction

Contextual bandit is a well-formulated abstraction of many important real-world problems, including content recommendation (Li et al., 2010; Wu et al., 2016), online advertising (Schwartz et al., 2017; Nuara et al., 2018), and mobile health (Lei et al., 2017; Tewari & Murphy, 2017). In such problems, an agent iteratively interacts with an environment to maximize its accumulated rewards over time. Its essence is sequential decision-making under uncertainty. Because the reward from the environment for a chosen action (also referred to as an arm in literature) under each context is stochastic, a no-regret learning algorithm needs to explore the problem space for improved reward estimation, i.e., learning the mapping from an arm and its context[1] to the expected reward.

Linear contextual bandit algorithms (Abbasi-Yadkori et al., 2011; Li et al., 2010), which assume the reward mapping is a linear function of the context vector, dominate the community's attention in the study of contextual bandits. Though theoretically sound and practically effective, their linear reward mapping assumption is incompetent to capture possible complex non-linear relations between the context vector and reward. This motivated the extended studies in parametric bandits, such as generalized linear bandits (Filippi et al., 2010; Faury et al., 2020) and kernelized bandits (Chowdhury & Gopalan, 2017; Krause & Ong, 2011). Recently, to unleash the power of representation learning, deep neural networks (DNN) have also been introduced to learn the underlying reward mapping directly. In (Zahavy & Mannor, 2019; Riquelme et al., 2018; Xu et al., 2020), a deep neural network is applied to provide a feature mapping, and exploration is performed at the last layer. NeuralUCB (Zhou et al., 2020) and NeuralTS (ZHANG et al., 2020) explore the entire neural network parameter space to obtain nearly optimal regret using the neural tangent kernel technique (Jacot et al., 2018). These neural contextual bandit algorithms significantly boosted empirical performance compared to their classical counterparts.

Nevertheless, a major practical concern of existing neural contextual bandit algorithms is their added computational cost when performing exploration. Take the recently developed NeuralUCB and NeuralTS for example. Their construction of the high-probability confidence set for model exploration

---

[1]When no ambiguity is invoked, we refer to the feature vector for an arm and its context as a context vector.

depends on the dimensionality of the network parameters and the learned context vectors' representations, which is often very large for DNNs. For instance, a performing neural bandit solution often has the number of parameters in the order of 100 thousands (if not less). It is prohibitively expensive to compute inverse of the induced covariance matrix on such a huge number of parameters, as required by their construction of confidence set. As a result, approximations, e.g., only using the diagonal of the covariance matrix (Zhou et al., 2020; ZHANG et al., 2020), are employed to make such algorithms operational in practice. But there is no theoretical guarantee for such diagonal approximations, which directly leads to the gap between the theoretical and empirical performance of the neural bandit algorithms.

In this work, to alleviate the computational overhead caused by the expensive exploration, we propose to eliminate explicit model exploration by learning a neural bandit model with perturbed rewards. At each round of model update, we inject pseudo noise generated from a zero-mean Gaussian distribution to the observed reward history. With the induced randomization, sufficient exploration is achieved when simply pulling the arm with the highest estimated reward. This brings in considerable advantage over existing neural bandit algorithms: no additional computational cost is needed to obtain no regret. We rigorously prove that with a high probability the algorithm obtains a $\widetilde{O}(\widetilde{d}\sqrt{T})$ regret, where $\widetilde{d}$ is the effective dimension of a neural tangent kernel matrix and $T$ is the number of rounds of interactions. This result recovers existing regret bounds for the linear setting where the effective dimension equals to the input feature dimension. Besides, our extensive empirical evaluations demonstrate the strong advantage in efficiency and effectiveness of our solution against a rich set of state-of-the-art contextual bandit solutions over both synthetic and real-world datasets.

## 2 RELATED WORK

Most recently, attempts have been made to incorporate DNNs with contextual bandit algorithms. Several existing work study neural-linear bandits (Riquelme et al., 2018; Zahavy & Mannor, 2019), where exploration is performed on the last layer of the DNN. Under the neural tangent kernel (NTK) framework (Jacot et al., 2018), NeuralUCB (Zhou et al., 2020) constructs confidence sets with DNN-based random feature mappings to perform upper confidence bound based exploration. NeuralTS (ZHANG et al., 2020) samples from the posterior distribution constructed with a similar technique. However, as the exploration is performed in the induced random feature space, the added computational overhead is prohibitively high, which makes such solutions impractical. The authors suggested diagonal approximations of the resulting covariance matrix, which however leave the promised theoretical guarantees of those algorithms up in the air.

Reward perturbation based exploration has been studied in a number of classical bandit models (Kveton et al., 2019a; 2020; 2019b). In a context-free $k$-armed bandit setting, Kveton et al. (2019b) proposed to estimate each arm's reward over a perturbed history and select the arm with the highest estimated reward at each round. Such an arm selection strategy is proved to be optimistic with a sufficiently high probability. Later this strategy has been extended to linear and generalized linear bandits (Kveton et al., 2019a; 2020). In (Kveton et al., 2020), the authors suggested its application to neural network models; but only some simple empirical evaluations were provided, without any theoretical justifications. Our work for the first time provides a rigorous regret analysis of neural contextual bandits with the perturbation-based exploration: a sublinear regret bound is still achievable in terms of the number of interactions between the agent and environment.

## 3 NEURAL BANDIT LEARNING WITH PERTURBED REWARDS

We study the problem of contextual bandit with finite $K$ arms, where each arm is associated with a $d$-dimensional context vector: $\mathbf{x}_i \in \mathbb{R}^d$ for $i \in [K]$. At each round $t \in [T]$, the agent needs to select one of the arms, denoted as $a_t$, and receives its reward $r_{a_t,t}$, which is generated as $r_{a_t,t} = h(\mathbf{x}_{a_t}) + \eta_t$. In particular, $h(\mathbf{x})$ represents the unknown underlying reward mapping function satisfying $0 \le h(\mathbf{x}) \le 1$ for any $\mathbf{x}$, and $\eta_t$ is an $R$-sub-Gaussian random variable that satisfies $\mathbb{E}[\exp(\mu\eta_t)] \le \exp[\mu^2 R^2]$ for all $\mu \ge 0$. The goal is to minimize pseudo regret over $T$ rounds:

$$R_T = \mathbb{E}\left[\sum_{t=1}^{T}\left(r_{a^*} - r_{t,a_t}\right)\right], \tag{3.1}$$

---

**Algorithm 1** Neural bandit with perturbed reward (NPR)

1: **Input:** Number of rounds $T$, regularization coefficient $\lambda$, perturbation parameter $\nu$, network width $m$, network depth $L$.
2: **Initialization:** $\boldsymbol{\theta}_0 = (\text{vec}(\mathbf{W}_1), \dots, \text{vec}(\mathbf{W}_L)] \in \mathbb{R}^p$ with Gaussian distribution: for $1 \le l \le L - 1$, $\mathbf{W}_l = (\mathbf{W}, 0; 0, \mathbf{W})$ with each entry of $\mathbf{W}$ sampled independently from $\mathcal{N}(0, 4/m)$; $\mathbf{W}_L = (\mathbf{w}^\top, -\mathbf{w}^\top)$ with each entry of $\mathbf{w}$ sampled independently from $\mathcal{N}(0, 2/m)$.
3: **for** $t = 1, \dots, T$ **do**
4:     **if** $t > K$ **then**
5:         Pull arm $a_t$ and receive reward $r_{t,a_t}$, where $a_t = \text{argmax}_{i \in [K]} f(\mathbf{x}_i, \boldsymbol{\theta}_{t-1})$.
6:         Generate $\{\gamma_s^t\}_{s \in [t]} \sim \mathcal{N}(0, \nu^2)$.
7:         Set $\boldsymbol{\theta}_t$ by the output of gradient descent for solving Eq (3.2).
8:     **else**
9:         Pull arm $a_k$.
10:     **end if**
11: **end for**

---

where $a^*$ is the optimal arm with the maximum expected reward.

To deal with the potential non-linearity of $h(\mathbf{x})$ and unleash the representation learning power of DNNs, we adopt a fully connected neural network $f(\mathbf{x}; \boldsymbol{\theta})$ to approximate $h(\mathbf{x})$: $f(\mathbf{x}; \boldsymbol{\theta}) = \sqrt{m}\mathbf{W}_L\phi\Big(\mathbf{W}_{L-1}\phi\big(\cdots\phi(\mathbf{W}_1\mathbf{x})\big)\Big)$, where $\phi(x) = \text{ReLU}(x)$, $\boldsymbol{\theta} = [\text{vec}(\mathbf{W}_1), \dots, \text{vec}(\mathbf{W}_L)] \in \mathbb{R}^p$ with $p = m + md + m^2(L-1)$, and depth $L \ge 2$. Each hidden layer is assumed to have the same width (i.e., $m$) for convenience in later analysis; but this does not affect the conclusion of our theoretical analysis.

Existing neural bandit solutions perform explicit exploration in the entire model space (Zhou et al., 2020; ZHANG et al., 2020; Zahavy & Mannor, 2019; Riquelme et al., 2018; Xu et al., 2020), which introduces prohibitive computational cost. And oftentimes the overhead is so high that approximation has to be employed (Zhou et al., 2020; ZHANG et al., 2020), which unfortunately breaks the theoretical promise of these algorithms. In our proposed model, to eliminate such explicit model exploration in neural bandit, a randomization strategy is introduced in the neural network update. We name the resulting solution as Neural bandit with Perturbed Rewards, or NPR in short. In NPR, at round $t$, the neural model is learned with the $t$ rewards perturbed with designed perturbations:

$$\min_{\boldsymbol{\theta}} \mathcal{L}(\boldsymbol{\theta}) = \sum_{s=1}^t \big(f(\mathbf{x}_{a_s}; \boldsymbol{\theta}) - (r_{s,a_s} + \gamma_s^t)\big)^2/2 + m\lambda\|\boldsymbol{\theta} - \boldsymbol{\theta}_0\|_2^2/2 \tag{3.2}$$

where $\{\gamma_s^t\}_{s=1}^t \sim \mathcal{N}(0, \sigma^2)$ are Gaussian random variables that are independently sampled in each round $t$, and $\sigma$ is a hyper-parameter that controls the strength of perturbation (and thus the exploration) in NPR. We use an $l_2$-regularized square loss for model estimation, where the regularization centers at the randomly initialization $\boldsymbol{\theta}_0$ with the trade-off parameter $\lambda$.

The detailed procedure of NPR is given in Algorithm 1. The algorithm starts by pulling all candidate arms once. This guarantees that for any arm, NPR is sufficiently optimistic compared to the true reward with respect to its approximation error (Lemma 4.4). When all the $K$ arms have been pulled once, the algorithm pulls the arm with the highest estimated reward, $a_t = \text{argmax}_i f(\mathbf{x}_i; \boldsymbol{\theta}_{t-1})$. Once received the feedback $r_{a_t,t}$, the model perturbs the entire reward history so far via a freshly sampled noise sequence $\{\gamma_s^t\}_{s=1}^t$, and updates the neural network by $\{(\mathbf{x}_{a_s}, r_{a_s,s} + \gamma_s^t)\}_{s=1}^t$ using gradient descent. In the regret analysis in Section 4, we prove that the variance from the added $\{\gamma_s^t\}_{s \in [t]}$ will lead to the necessary optimism for exploration (Abbasi-Yadkori et al., 2011). We adopt gradient descent for analysis convenience, while stochastic gradient descent can also be used to solve the optimization problem with a similar theoretical guarantee based on recent works (Allen-Zhu et al., 2019; Zou et al., 2019).

Compared with existing neural contextual bandit algorithms (Zhou et al., 2020; ZHANG et al., 2020; Zahavy & Mannor, 2019; Riquelme et al., 2018; Xu et al., 2020), NPR does not need any added computation for model exploration, besides the regular neural network update. This greatly alleviates the overhead for the computation resources (both space and time) and makes NPR sufficiently general to be applied for practical problems. More importantly, our theoretical analysis directly corresponds to its actual behavior when applied, as no approximation is needed.

## 4 REGRET ANALYSIS

In this section, we provide finite time regret analysis of NPR, where the time horizon $T$ is set beforehand. The theoretical analysis is built on the recent studies about the generalization of DNN models (Cao & Gu, 2020; 2019; Chen et al., 2020; Daniely, 2017; Arora et al., 2019), which illustrate that with (stochastic) gradient descent, the learned parameters of a DNN locate in a particular regime with the generalization error being characterized by the best function in the corresponding neural tangent kernel (NTK) space (Jacot et al., 2018). We leave the background of NTK in the appendix and focus on the key steps of our proof in this section.

The analysis starts with the following lemma for the set of context vectors $\{\mathbf{x}_i\}_{i=1}^K$.

**Lemma 4.1.** There exists a positive constant $\bar{C}$ such that for any $\delta \in (0, 1)$, with probability at least $1 - \delta$, when $m \geq \bar{C}K^4L^6 \log(K^2L/\delta)/\lambda_0^4$, there exists a $\boldsymbol{\theta}^* \in \mathbb{R}^p$, for all $i \in [K]$,

$$h(\mathbf{x}_i) = \langle \mathbf{g}(\mathbf{x}_i; \boldsymbol{\theta}_0), \boldsymbol{\theta}^* - \boldsymbol{\theta}_0 \rangle, \quad \sqrt{m}\|\boldsymbol{\theta}^* - \boldsymbol{\theta}_0\|_2 \leq \sqrt{2\mathbf{h}^\top \mathbf{H}^{-1}\mathbf{h}}, \tag{4.1}$$

where $\mathbf{H}$ is the NTK matrix defined on the context set and $\mathbf{h} = (h(\mathbf{x}_1), \ldots, h(\mathbf{x}_K))$. Details of $\mathbf{H}$ can be found in the appendix.

This lemma suggests that with a satisfied neural network width $m$, with high probability, the underlying reward mapping function can be approximated by a linear function over $\mathbf{g}(\mathbf{x}^i; \boldsymbol{\theta}_0)$, parameterized by $\boldsymbol{\theta}^* - \boldsymbol{\theta}_0$, where $\mathbf{g}(\mathbf{x}^i; \boldsymbol{\theta}_0) = \nabla_{\boldsymbol{\theta}} f(\mathbf{x}; \boldsymbol{\theta}_0) \in \mathbb{R}^d$ is the gradient of the initial neural network. We also define the covariance matrix $\mathbf{A}_t$ at round $t$ as,

$$\mathbf{A}_t = \sum_{s=1}^{t-1} \mathbf{g}(\mathbf{x}_{s,a_s}; \boldsymbol{\theta}_0)\mathbf{g}(\mathbf{x}_{s,a_s}; \boldsymbol{\theta}_0)^\top/m + \lambda\mathbf{I} \tag{4.2}$$

where $\lambda > 0$ is the $l_2$ regularization parameter in Eq (3.2). Discussed before, there exist two kinds of noises in the training samples: the observation noise and the perturbation noise. To analyze the effect of each noise in model estimation, two auxiliary least square solutions are introduced, $\mathbf{A}_t^{-1}\bar{\mathbf{b}}_t$ and $\mathbf{A}_t^{-1}\mathbf{b}_t$ with

$$\bar{\mathbf{b}}_t = \sum_{s=1}^{t-1}(r_{s,a_s} + \mathbb{E}[\gamma_s^{t-1}])\mathbf{g}(\mathbf{x}_{s,a_s}; \boldsymbol{\theta}_0)/\sqrt{m}, \quad \mathbf{b}_t = \sum_{s=1}^{t-1}(r_{s,a_s} + \gamma_s^{t-1})\mathbf{g}(\mathbf{x}_{s,a_s}; \boldsymbol{\theta}_0)/\sqrt{m}.$$

These two auxiliary solutions isolate the induced deviations: the first least square solution only contains the deviation caused by observation noise $\{\eta_s\}_{s=1}^{t-1}$; and the second least square solution has an extra deviation caused by $\{\gamma_s^{t-1}\}_{s=1}^{t-1}$.

To analysis the estimation error of the neural network, we first define the following events. At round $t$, we have event $E_{t,1}$ defined as:

$$E_{t,1} = \left\{\forall i \in [K], |\langle \mathbf{g}(\mathbf{x}_{t,i}; \boldsymbol{\theta}_0), \mathbf{A}_t^{-1}\bar{\mathbf{b}}_t/\sqrt{m}\rangle - h(\mathbf{x}_{t,i})| \leq \alpha_t\|\mathbf{g}(\mathbf{x}_{t,i}; \boldsymbol{\theta}_0)\|_{\mathbf{A}_t^{-1}}\right\}.$$

According to Lemma 4.1 and the definition of $\bar{\mathbf{b}}_t$, under event $E_{t,1}$, the deviation caused by the observation noise $\{\eta_s\}_{s=1}^{t-1}$ can be controlled by $\alpha_t$. We also need event $E_{t,2}$ at round $t$ defined as

$$E_{t,2} = \{\forall i \in [K] : |f(\mathbf{x}_{t,i}; \boldsymbol{\theta}_{t-1}) - \langle \mathbf{g}(\mathbf{x}_{t,i}; \boldsymbol{\theta}_0), \mathbf{A}_t^{-1}\bar{\mathbf{b}}_t/\sqrt{m}\rangle| \leq \epsilon(m) + \beta_t\|\mathbf{g}(\mathbf{x}_{t,i}; \boldsymbol{\theta}_0)\|_{\mathbf{A}_t^{-1}}\},$$

where $\epsilon(m)$ is defined as

$$\begin{aligned}\epsilon(m) =&C_{\epsilon,1}m^{-1/6}T^{2/3}\lambda^{-2/3}L^3\sqrt{\log m} + C_{\epsilon,2}(1 - \eta m\lambda)^J\sqrt{TL/\lambda} \\ &+ C_{\epsilon,3}m^{-1/6}T^{5/3}\lambda^{-5/3}L^4\sqrt{\log m}(1 + \sqrt{T/\lambda}),\end{aligned}$$

with constants $\{C_{\epsilon,i}\}_{i=1}^3$. Event $E_{t,2}$ considers the situation that the deviation caused by the added noise $\{\gamma_s^{t-1}\}_{s=1}^{t-1}$ is under control with parameter $\beta_t$ at round $t$, and the approximation error $\epsilon(m)$ is carefully tracked by properly setting the neural network width $m$.

The following lemmas show that given the history $\mathcal{F}_{t-1} = \sigma\{\mathbf{x}_1, r_1, \mathbf{x}_2, r_2, ..., \mathbf{x}_{t-1}, r_{t-1}, \mathbf{x}_t\}$, which is $\sigma$-algebra generated by the previously pulled arms and the observed rewards, with proper setting, events $E_{t,1}$ and $E_{t,2}$ happen with a high probability.

**Lemma 4.2.** For any $t \in [T]$, $\lambda > 0$ and $\delta > 0$, with $\alpha_t = \sqrt{R^2 \log \big( \det(\mathbf{A}_t)/(\delta^2 \det(\lambda \mathbf{I})) \big)} + \sqrt{\lambda} S$, we have $\mathbb{P}(E_{t,1}|\mathcal{F}_{t-1}) \geq 1 - \delta$.

In particular, $S$ denotes the upper bound of $\sqrt{2\mathbf{h}^\top \mathbf{H}^{-1} \mathbf{h}}$, which is a constant if the reward function $h(\cdot)$ belongs to the RKHS space induced by the neural tangent kernel.

**Lemma 4.3.** There exist positive constants $\{C_i\}_{i=1}^3$, such that with step size and the neural network satisfy that $\eta = C_1(m\lambda + mLT)^{-1}$, $m \geq C_2 \sqrt{\lambda} L^{-3/2} [\log(TKL^2/\delta)]^{3/2}$, and $m[\log m]^{-3} \geq C_3 \max\{TL^{12}\lambda^{-1}, T^7\lambda^{-8}L^{18}(\lambda + LT)^6, L^{21}T^7\lambda^{-7}(1 + \sqrt{T/\lambda})^6\}$, given history $\mathcal{F}_{t-1}$, by choosing $\beta_t = \sigma\sqrt{4\log t + 2\log K}$, we have $\mathbb{P}(E_{t,2}|\mathcal{F}_{t-1}) \geq 1 - t^{-2}$.

We leave the detailed proof in the appendix. According to Lemma 4.2 and Lemma 4.3, with more observations become available, the neural network based reward estimator $f(\mathbf{x}; \boldsymbol{\theta}_t)$ will gradually converge to the true expected reward $h(\mathbf{x})$. In the meantime, the variance of the added perturbation leads to reward overestimation, which will compensate the potential underestimation caused by the observation noise and the approximation error. The following lemma describes this anti-concentration behavior of the added noise $\{\gamma_s^{t-1}\}_{s=1}^{t-1}$.

**Lemma 4.4.** For any $t \in [T]$, with $\sigma = \alpha_t(1 - \lambda\lambda_K^{-1}(\mathbf{A}_K))^{-1/2}$, $\alpha_t$ defined in Lemma 4.2, $\epsilon(m)$ defined in Lemma 4.3, given history $\mathcal{F}_{t-1}$, we have $\mathbb{P}(E_{t,3}) = \mathbb{P}(f(\mathbf{x}_{t,i}; \boldsymbol{\theta}_{t-1}) \geq h(\mathbf{x}_{t,i}) - \epsilon(m)|\mathcal{F}_{t-1}, E_{t,1}) \geq (4e\sqrt{\pi})^{-1}$, where $\mathbf{A}_K$ is the covariance matrix constructed after the initial $K$-round arm pulling (line 8 to 10 in Algorithm 1) by Eq (4.2), and $\lambda_K(\mathbf{A}_K)$ represents the $K$-th largest eigenvalue of matrix $\mathbf{A}_K$.

With $\alpha_t$ defined in Lemma 4.2, and $\epsilon(m)$ and $\beta_t$ defined in Lemma 4.3, we divide the arms into two groups. More specifically, we define the set of sufficiently sampled arms in round $t$ as:

$$\Omega_t = \{\forall i \in [K] : 2\epsilon(m) + (\alpha_t + \beta_t)\|\mathbf{g}(\mathbf{x}_{t,i}; \boldsymbol{\theta}_0)\|_{\mathbf{A}_t^{-1}} \leq h(\mathbf{x}_{t,a^*}) - h(\mathbf{x}_{t,i})\}.$$

Accordingly, the set of undersampled arms is defined as $\bar{\Omega}_t = [K] \setminus \Omega_t$. Note that by this definition, the best arm $a^*$ belongs to $\bar{\Omega}_t$. In the following lemma, we show that at any time step $t > K$, the expected one-step regret is upper bounded in terms of the regret due to playing an undersampled arm $a_t \in \bar{\Omega}_t$.

**Lemma 4.5.** With $m$, $\eta$, $\sigma$ satisfying the conditions in Lemma 4.2, 4.3, 4.4, and $\epsilon(m)$ defined in Lemma 4.3, with probability at least $1 - \delta$ the one step regret of NPR is upper bounded,

$$\mathbb{E}[h(\mathbf{x}_{t,a_t^*}) - h(\mathbf{x}_{t,a_t})] \leq \mathbb{P}(\bar{E}_{t,2}) + 4\epsilon(m) + (\beta_t + \alpha_t)\big(1 + 2/\mathbb{P}(a_t \in \bar{\Omega}_t)\big)\|\mathbf{g}(\mathbf{x}_{t,a_t}; \boldsymbol{\theta}_0)/\sqrt{m}\|_{\mathbf{A}_t^{-1}}.$$

Furthermore, $\mathbb{P}(a_t \in \bar{\Omega}_t) \geq \mathbb{P}_t(E_{t,3}) - \mathbb{P}_t(\bar{E}_{t,2})$.

To bound the cumulative regret of NPR, we also need the following lemma.

**Lemma 4.6.** With $\tilde{d}$ as the effective dimension of the NTK matrix $\mathbf{H}$, we have

$$\sum_{t=1}^T \min\left\{\|\mathbf{g}(\mathbf{x}_{t,a_t}; \boldsymbol{\theta}_0)/\sqrt{m}\|_{\mathbf{A}_{t-1}^{-1}}^2, 1\right\} \leq 2(\tilde{d}\log(1 + TK/\lambda) + 1)$$

The effective dimension is first proposed in (Valko et al., 2013) to analyze the kernelized contextual bandit, which roughly measures the number of directions in the kernel space where the data mostly lies. The detailed definiton of $\tilde{d}$ can be found in Definition B.3 in the appendix. Finally, with proper neural network setup, we provide an $m$-independent regret upper bound.

**Theorem 4.7.** Let $\tilde{d}$ be the effective dimension of the NTK kernel matrix $\mathbf{H}$. There exist positive constants $C_1$ and $C_r$, such that for any $\delta \in (0, 1)$, when network structure and the step size for model update satisfy $m \geq \text{poly}(T, L, K, \lambda^{-1}, S^{-1}, \log(1/\delta))$, $\eta = C_1(mTL + m\lambda)^{-1}$ and $\lambda \geq \max\{1, S^{-2}\}$, with probability at least $1 - \delta$, the cumulative regret of NPR satisfies

$$R_T \leq \Gamma\Big(R\sqrt{\tilde{d}\log(1 + TK/\lambda) + 2 - 2\log\delta} + \sqrt{\lambda}S\Big) \cdot \sqrt{2\tilde{d}T\log(1 + TK/\lambda) + 2} + 7 + K \tag{4.3}$$

where $\Gamma = 44e\sqrt{\pi}(1 + \sqrt{(4\log T + 2\log K)/(1 - \lambda\lambda_K^{-1}(\mathbf{A}_K))})$.

*Proof.* By Lemma 4.2, $E_{t,1}$ holds for all $t \in [T]$ with probability at least $1 - \delta$. Therefore, with probability at least $1 - \delta$, we have

$$R_T = \sum_{t=1}^{T} \mathbb{E}[(h(\mathbf{x}_{t,a_t^*}) - h(\mathbf{x}_{t,a_t}))\mathbb{1}\{E_{t,1}\}] \leq \sum_{t=K+1}^{T} \mathbb{E}[(h(\mathbf{x}_{t,a_t^*}) - h(\mathbf{x}_{t,a_t}))\mathbb{1}\{E_{t,1}\}] + K$$

$$\leq 4T\epsilon(m) + \sum_{t=1}^{T} \mathbb{P}_t(\bar{E}_{t,2}) + (\beta_t + \alpha_t)(1 + \frac{2}{\mathbb{P}_t(E_{t,3}) - \mathbb{P}_t(\bar{E}_{t,2})})\|\mathbf{g}(\mathbf{x}_{t,a_t};\boldsymbol{\theta}_0)/\sqrt{m}\|_{\mathbf{A}_t^{-1}} + K.$$

The second inequality is due to the analysis of one-step regret in Lemma 4.5. According to Lemma 4.3, by choosing $\delta = \sigma c_t$, with $c_t = \sqrt{4 \log t + 2 \log K}$, $\mathbb{P}_t(\bar{E}_{t,2}) \leq t^{-2}$. Hence, $\sum_{t=1}^{T} \mathbb{P}_t(\bar{E}_{t,2}) \leq \pi^2/6$. Based on Lemma 4.4, with $\sigma = \alpha_t(1 - \lambda\lambda_K^{-1}(\mathbf{A}_K))^{-1/2}$, $\mathbb{P}_t(E_{t,3}) \geq \frac{1}{4\mathbf{e}\sqrt{\pi}}$. Therefore, $1 + 2/(\mathbb{P}_t(E_{t,3}) - \mathbb{P}_t(\bar{E}_{t,2})) \leq 44\mathbf{e}\sqrt{\pi}$. By chaining all the inequalities, we have the cumulative regret of NPR upper bounded as,

$$R_T \leq \frac{\pi^2}{3} + 4T\epsilon(m) + \sum_{t=1}^{T}(\beta_t + \alpha_t)(1 + \frac{2}{\mathbb{P}_t(E_{t,3}) - \mathbb{P}_t(\bar{E}_{t,2})})\|\mathbf{g}(\mathbf{x}_{t,a_t};\boldsymbol{\theta}_0)/\sqrt{m}\|_{\mathbf{A}_t^{-1}} + K$$

$$\leq \Gamma\Big(R\sqrt{\widetilde{d}\log(1 + TK/\lambda) + 2 - 2\log\delta} + \sqrt{\lambda}S\Big) \cdot \sqrt{2\widetilde{d}T\log(1 + TK/\lambda) + 2} + \frac{\pi^2}{3} + K$$

$$+ C_{\epsilon,1}m^{-1/6}T^{5/3}\lambda^{-2/3}L^3\sqrt{\log m} + C_{\epsilon,2}(1 - \eta m\lambda)^J T\sqrt{TL/\lambda}$$

$$+ C_{\epsilon,3}m^{-1/6}T^{8/3}\lambda^{-5/3}L^4\sqrt{\log m}(1 + \sqrt{T/\lambda})$$

where $\Gamma = 44\mathbf{e}\sqrt{\pi}(1 + \sqrt{(4\log T + 2\log K)/(1 - \lambda\lambda_K^{-1}(\mathbf{A}_K))})$. The second inequality is due to Lemma 4.6, and the bounds of events $\bar{E}_{t,2}$ and $E_{t,3}$. By setting $\eta = c(m\lambda + mLT)^{-1}$, and $J = (1 + LT/\lambda)(\log(C_{\epsilon,2}) + \log T\sqrt{TL/\lambda})/c$, $C_{\epsilon,2}(1 - \eta m\lambda)^J T\sqrt{TL/\lambda} \leq 1$. Then by choosing the satisfied $m$, $C_{\epsilon,1}m^{-1/6}T^{5/3}\lambda^{-2/3}L^3\sqrt{\log m} + C_{\epsilon,3}m^{-1/6}T^{8/3}\lambda^{-5/3}L^4\sqrt{\log m}(1 + \sqrt{T/\lambda}) \leq 2/3$. $R_T$ can be further bounded by $R_T \leq \Gamma\Big(R\sqrt{\widetilde{d}\log(1 + TK/\lambda) + 2 - 2\log\delta} + \sqrt{\lambda}S\Big) \cdot \sqrt{2\widetilde{d}T\log(1 + TK/\lambda) + 2} + 7 + K$

This completes the proof. □

## 5 EXPERIMENTS

In this section, we empirically evaluate the proposed neural bandit algorithm NPR against several state-of-the-art baselines, including: linear and neural UCB (LinUCB (Abbasi-Yadkori et al., 2011) and NeuralUCB (Zhou et al., 2020)), linear and neural Thompson Sampling (LinTS (Agrawal & Goyal, 2013) and NeuralTS (ZHANG et al., 2020)), perturbed reward for linear bandits (LinFPL) (Kveton et al., 2019a). To demonstrate the real impact of using diagonal approximation in confidence set construction, we also include NeuralUCB and NeuralTS with diagonal approximation as suggested in their original papers for comparisons.

We evaluated all the algorithms on 1) synthetic dataset via simulation, 2) six $K$-class classification datasets from UCI machine learning repository (Beygelzimer et al., 2011), and 3) two real-world datasets extracted from the social bookmarking web service Delicious and music streaming service LastFM (Wu et al., 2016). We implemented all the algorithms in PyTorch and performed all the experiments on a server equipped with Intel Xeon Gold 6230 2.10GHz CPU, 128G RAM, four NVIDIA GeForce RTX 2080Ti graphical cards.

### 5.1 EXPERIMENT ON SYNTHETIC DATASET

In our simulation, we first generate a size-$K$ arm pool, in which each arm $i$ is associated with a $d$-dimensional feature vector $\mathbf{x}_i$. The contextual vectors are sampled uniformly at random from a unit ball. We generate the rewards via the following two nonlinear functions:

$$h_1(\mathbf{x}) = 10^{-2}(\mathbf{x}^\top \Sigma \Sigma^\top \mathbf{x}), \quad h_2(\mathbf{x}) = \exp(-10(\mathbf{x}^\top \boldsymbol{\theta})^2),$$

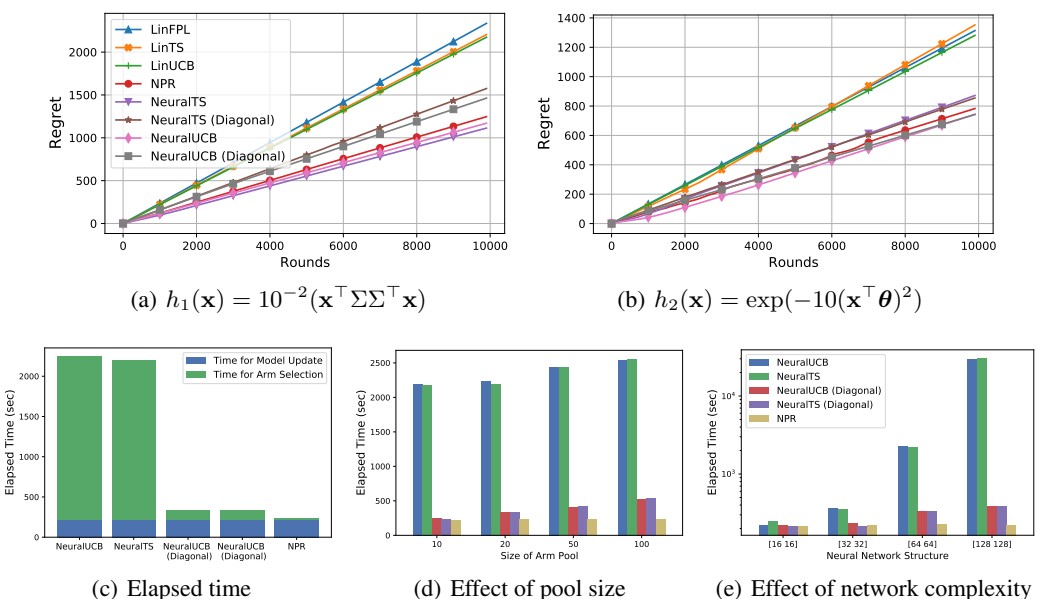

Figure 1: Empirical results of regret and time consumption on synthetic dataset.

where each element of $\Sigma \in \mathbb{R}^{d \times d}$ is randomly sampled from $\mathcal{N}(0, 1)$, and $\boldsymbol{\theta}$ is randomly sampled from a unit ball. At each round $t$, only a subset of $k$ arms out of the total $K$ arms are sampled without replacement and disclosed to all algorithms for selection. The ground-truth reward $r_a$ is corrupted by Gaussian noise $\eta = \mathcal{N}(0, \xi^2)$ before feeding back to the bandit algorithms. We fixed the feature dimension $d = 50$ and the pool size $K = 100$ with $k = 20$. $\xi$ was set to 0.1 for both $h_1$ and $h_2$.

Cumulative regret is used to compare the performance of different algorithms. Here the best arm is defined as the one with the highest expected reward in the presented $k$ candidate arms. All the algorithms were executed up to 10000 rounds in simulation, and the averaged results over 10 runs are reported in Figure 1(a) to 1(e). For the neural bandit algorithms, we adopted a 3-layer neural network with $m = 64$ units in each hidden layer. We did a grid search on the first 1000 rounds for regularization parameter $\lambda$ over $\{10^{-i}\}_{i=0}^4$, and step size $\eta$ over $\{10^{-i}\}_{i=0}^3$. We also searched for the concentration parameter $\delta$ so that the exploration parameter $\nu$ is equivalently searched over $\{10^{-i}\}_{i=0}^4$. The best set of hyper-parameters for each algorithm are applied for the full simulation over 10000 rounds, i.e., only that trial will be continued for the rest of 9000 rounds.

In Figure 1(a) and 1(b), we can observe that linear bandit algorithms clearly failed to learn non-linear mapping and suffered much higher regret compared to the neural algorithms. NPR achieved similar performance as NeuralUCB and NeuralTS, which shows the effectiveness of our randomized exploration strategy in neural bandit learning.

Figure 1(c) shows the average running time for all the neural bandit models in this experiment. With the same network structure, the time spent for model update was similar across different neural bandit models. But we can clearly observe that NeuralUCB and NeuralTS took much more time in arm selection. When choosing an arm, NeuralUCB and NeuralTS, and their corresponding versions with diagonal approximation, need to repeatedly compute: 1) the inverse of the covariance matrix based on the pulled arms in history; and 2) the confidence set of their reward estimation on each candidate arm. The size of the covariance matrix depends on the number of parameters in the neural network (e.g., 7460-by-7460 in this experiment). With diagonal approximation, the computational cost is significantly reduced. But, as mentioned before, there is no theoretical guarantee about the approximation's impact on the model's performance. Intuitively, the impact depends on how the full covariance matrix concentrates on its diagonal, e.g., whether the random feature mappings of the network on the pulled arms are correlated among their dimensions. Unfortunately, this cannot be determined in advance. Figure 1(a) and 1(b) show that the diagonal approximation leads to a significant performance drop. This confirms our concern of such an approximation. In contrast, as shown in Figure 1(c), there is no extra computation overhead in NPR; in the meantime, it provides comparable performance to NeuralUCB and NeuralTS with a full covariance matrix.

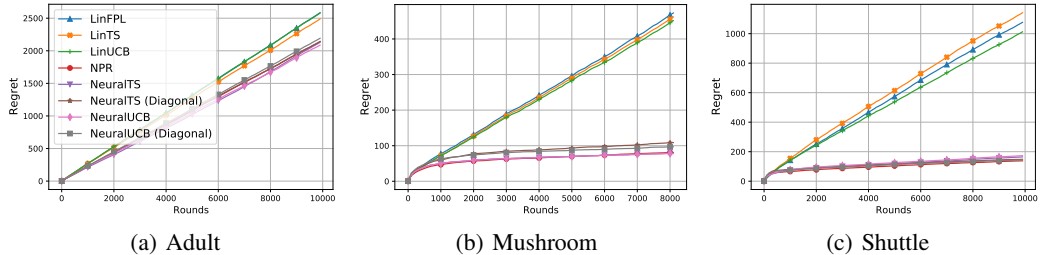

Figure 2: Comparison of NPR and the baselines on the UCI datasets.

To further investigate the efficiency advantage of NPR, we reported the total running time for the neural bandit algorithms under different sizes of the arm pool and different structures of the neural network under the same simulation setting. With the same neural network structure, models with different sizes of candidate arms should share the same running time for model update. Hence, Figure 1(d) reports how the size of arm pool affects the time spent on arm selection. As we can find, with more candidate arms, NeuralUCB, NeuralTS, and their corresponding diagonal approximations spent more time on arm selection, or more specifically on constructing the confidence set of the reward estimation for each arm. With a fixed size arm pool, the complexity of the neural network will strongly affect the time for computing the inverse of the covariance matrix. In Figure 1(e), the x-axis shows the structure for the hidden layers. The running time for NeuralUCB and NeuralTS significantly increased with enlarged neural networks. The running time of NPR stayed stable, across varying sizes of arm pools and the network complexity. This shows the strong advantage of NPR in efficiency and effectiveness, especially for large-scale problems.

## 5.2 EXPERIMENT ON CLASSIFICATION DATASETS

Following the setting in (Zhou et al., 2020), we evaluated the bandit algorithms on $K$-class classification problem with six public benchmark datasets, including `adult`, `mushroom`, and `shuttle` from UCI Machine Learning Repository (Dua & Graff, 2017). Due to space limit, we report the results on these three datasets in this section, and leave the other three, `letter`, `covertype`, and `MagicTelescope` in the appendix. We adopted the disjoint model (Li et al., 2010) to build the context vectors for candidate arms: given an input instance with feature $\mathbf{x} \in \mathbb{R}^d$ of a $k$-class classification problem, we constructed the context features for $k$ candidate arms as: $\mathbf{x}_1 = (\mathbf{x}, \mathbf{0}, \ldots, \mathbf{0}), \ldots, \mathbf{x}_k = (\mathbf{0}, \ldots, \mathbf{0}, \mathbf{x}) \in \mathbb{R}^{d \times k}$. The model receives reward 1 if it selects the correct class by pulling the corresponding arm, otherwise 0. The cumulative regret measures the total mistakes made by the model over $T$ rounds. Similar to the experiment in synthetic datasets, we adopted a 3-layer neural network with $m = 64$ units in each hidden layer. We used the same grid search setting as in our experiments on synthetic datasets. Figure 2 shows the averaged results across 10 runs for 10000 rounds, except for `mushroom` which only contains 8124 data instances.

There are several key observations in Figure 2. First, the improvement of applying a neural bandit model depends on the nonlinearity of the dataset. For example, the performance on `mushoom` and `shuttle` was highly boosted by the neural models, while the improvement on `adult` was limited. Second, the impact of the diagonal approximation for NeuralUCB and NeuralTS also depends on the dataset. It did not hurt their performance too much on `shuttle`, while using the full covariance matrix led to much better performance on `Mushroom`. Overall, NPR showed consistently better or at least comparable performance to the best neural bandit baselines in all six datasets. Such robust performance and high efficiency make it more advantageous in practice.

## 5.3 EXPERIMENT ON LASTFM & DELICIOUS

The LastFM dataset was extracted from the music streaming service website Last.fm, and the Delicious data set was extracted from the social bookmark sharing service website Delicious (Wu et al., 2016). In particular, the LastFM dataset contains 1,892 users and 17,632 items (artists). We treat the "listened artists" in each user as positive feedback. The Delicious dataset contains 1,861 users and 69,226 items (URLs). We treat the bookmarked URLs for each user as positive feedback. Following the standard setting (Cesa-Bianchi et al., 2013), we pre-processed these two datasets. For the item

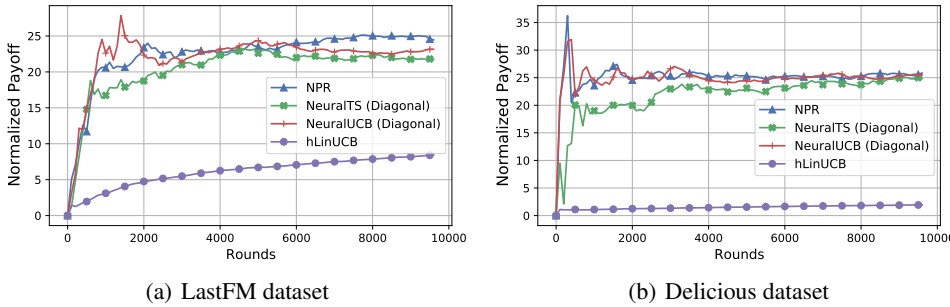

(a) LastFM dataset          (b) Delicious dataset

Figure 3: Comparisons of normalized rewards on LastFM & Delicious datasets.

features, we used all the tags associated with each item to create its TF-IDF feature vector. Then PCA was applied to reduce the dimensionality of the features. In both datasets, we used the first 25 principle components to construct the item feature vectors. On the user side, we created user features by running node2vec (Grover & Leskovec, 2016) on the user relation graph extracted from the social network provided by the datasets. Each of the users was represented with a 50-dimensional feature vector. We generated the candidate pool as follows: we fixed the size of candidate arm pool to $k = 25$ for each round; for each user, we picked one item from those non-zero payoff items according to the complete observations in the dataset, and randomly picked the other 24 from those zero-payoff items. We constructed the context vectors by concatenating the user and item feature vectors. In our experiment, a 3-layer neural network with $m = 128$ units in each hidden layer was applied to learn the bandit model. And the same gird search was performed to find the best set of hyper-parameters. We performed the experiment with 10000 rounds. Because of the high demand of GPU memory for the matrix inverse computation, we only compared NPR to NeuralUCB and NeuralTS with the diagonal approximation.

We compared the cumulative reward from each algorithm in Figure 3. Besides the neural contextual bandit algorithms, we also include the best reported model on these two datasets, hLinUCB (Wang et al., 2016) for comparison. To improve visibility, we normalized the cumulative reward by a random strategy's cumulative reward (Wu et al., 2016), where the arm was randomly sampled from the candidate pool. All the neural models showed the power to learn the potential non-linear reward mapping on these two real-world datasets. NPR showed its advantage over the baselines (especially on LastFM). As NPR has no additional requirement on the computation resource (e.g., no need for the expensive matrix operation as in the baselines), we also experienced much shorter running in NPR than the other two neural bandit models.

## 6 CONCLUSION

Existing neural contextual bandit algorithms sacrifice computation efficiency for effective exploration in the entire neural network parameter space. To eliminate the computational overhead caused by such an expensive operation, we appealed to randomization-based exploration and obtained the same level of learning outcome. No extra computational cost or resources are needed besides the regular neural network update. We proved that the proposed solution obtains $\widetilde{\mathbf{O}}(\widetilde{d}\sqrt{T})$ regret for $T$ rounds of interactions between the system and user. Extensive empirical results on both synthetic and real-world datasets support our theoretical analysis, and demonstrate the strong advantage of our model in efficiency and effectiveness, which suggest the potential of deploying powerful neural bandit model online in practice.

Our empirical efficiency analysis shows that the key bottleneck of learning a neural model online is at the model update step. Our current regret analysis depends on (stochastic) gradient descent over the entire training set for model update in each round, which is prohibitively expensive. We would like to investigate the possibility of more efficient model update, e.g., online stochastic gradient descent or continual learning, and the corresponding effect on model convergence and regret analysis. Currently, our analysis focuses on the $K$-armed setting, and it is important to extend our solution and analysis to infinitely many arms in our future work.

## ACKNOWLEDGEMENTS

We thank the anonymous reviewers for their helpful comments. YJ and HW are partially supported by the National Science Foundation under grant IIS-2128019 and IIS-1553568. WZ, DZ and QG are partially supported by the National Science Foundation CAREER Award 1906169 and IIS-1904183. The views and conclusions contained in this paper are those of the authors and should not be interpreted as representing any funding agencies.

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

## A  ADDITIONAL EXPERIMENTS

### A.1  EXPERIMENTS ON SYNTHETIC DATASETS

#### A.1.1  REWARD PREDICTION WITH MORE COMPLICATED NEURAL NETWORKS

In the main paper, to compare NPR with the existing neural bandit baselines, we chose a relatively small neural network, e.g., $m = 64$, so that NeuralUCB/NeuralTS can be executed with a full co-variance matrix. To better capture the non-linearity in reward generation, we increased the width of the neural network to $m = 128$, and report the results in Figure 4. All other experiment settings (e.g., underlying reward generation function) are the same as reported in the main paper. And to provide a more comprehensive picture, we also reported the variance of each algorithm's performance in this figure.

With the increased network width and limited by the GPU memory, NeuralUCB and NeuralTS can only be executed with the diagonal matrix of the covariance matrix when computing their required "exploration bonus term". In Figure 4, we can observe 1) all neural bandit algorithms obtained the expected sublinear regret; and 2) NPR performed better than NeuralUCB and NeuralTS, because of the diagonal approximation that we have to use on these two baselines. This again strongly suggested the advantage of NPR: the approximation due to the computational complexity in NeuralUCB and NeuralTS limits their practical effectiveness, though they have the same theoretical regret bound as NPR's. 3) FALCON+ shows better performance in the initial rounds, while with more iterations, its regret is higher than the neural bandit baselines' and our NPR model'. Compared to the neural bandit models, FALCON+ has much slower convergence, e.g, the slope of its regret curve did not decrease as fast as other models' over the course of interactions. After a careful investigation, we found this is caused by its scheduled update: because less and less updates can be performed in the later rounds, FALCON+ still selected the suboptimal arms quite often. Another important observation is that FALCON+ has a very high variance in its performance: in some epochs, it could keep sampling the suboptimal arms. This is again caused by its lazy update: in some epochs of some trials, its estimation could be biased by large reward noise, which will directly cost it miss the right arm in almost the entire next epoch.

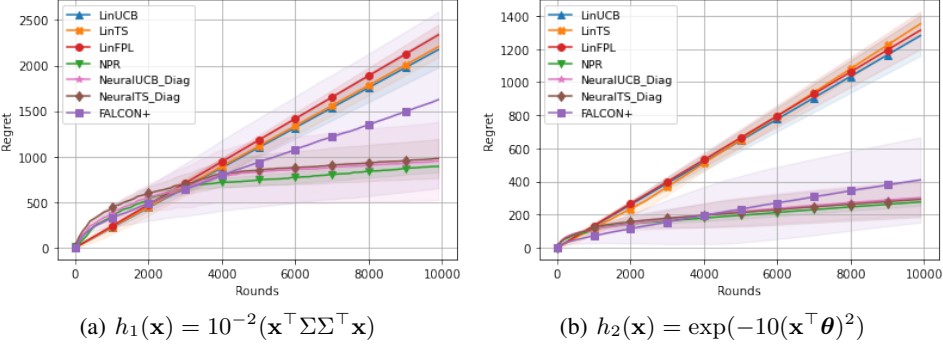

(a) $h_1(\mathbf{x}) = 10^{-2}(\mathbf{x}^\top \Sigma \Sigma^\top \mathbf{x})$  (b) $h_2(\mathbf{x}) = \exp(-10(\mathbf{x}^\top \boldsymbol{\theta})^2)$

Figure 4: Comparison of NPR and the baselines on the synthetic dataset with $m = 128$

#### A.1.2  COMPARISONS UNDER DIFFERENT VALUES OF $k$

For the experiment in the main paper, at each round only $k = 20$ arms are sampled from the $K = 100$ arm pool and disclosed to all the algorithms for selection. It is to test a more general setting in practice, e.g., different recommendation candidates appear across rounds. In the meantime, such setting also introduces more practical difficulties as only a subset of arms are revealed to the agent to learn from each time. In this experiment, we report the results of the neural bandit models with

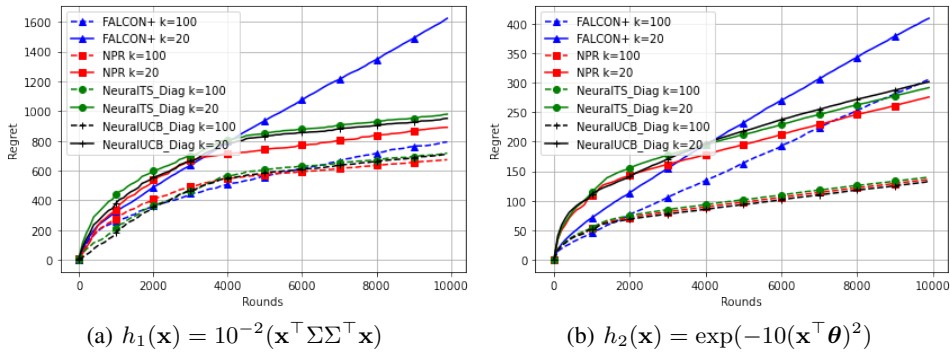

(a) $h_1(\mathbf{x}) = 10^{-2}(\mathbf{x}^\top \Sigma\Sigma^\top \mathbf{x})$          (b) $h_2(\mathbf{x}) = \exp(-10(\mathbf{x}^\top \boldsymbol{\theta})^2)$

Figure 5: Comparison of NPR and the baselines on the synthetic dataset with different $k$.

different value $k$, e.g., the number of arms disclosed each round. We followed the setting in Section A.1.1 to set the network width $m = 128$.

Figure 5 demonstrates that for both datasets, revealing all the candidate arms, e.g., $k = 100$, will considerably reduce the regret compared to only revealing a subset of the arms. But in both settings, all bandit algorithms, including NPR, can obtain desired performance (i.e., sublinear regret). And again, NPR still obtained more satisfactory performance as it is not subject to the added computational complexity in NeuralUCB and NeuralTS. The FALCON+ baseline performed much better when $k = 20$ than $k = 100$, as it has less chance to select the suboptimal arms in each epoch, which is expected based on its polynomial dependence on the number of arms $K$. On the contrary, NPR only has a logarithmic dependence on $K$, whose advantage was validated in this experiment.

## A.2 EXPERIMENTS ON CLASSIFICATION DATASETS

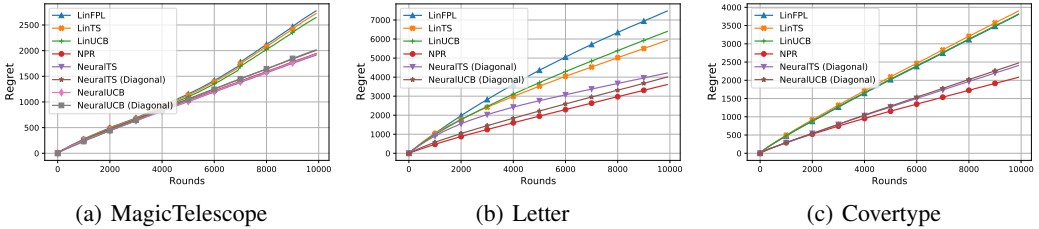

(a) MagicTelescope          (b) Letter          (c) Covertype

Figure 6: Comparison of NPR and the baselines on the UCI datasets.

Table 1: Dataset statistics

| DATASET | ADULT | MUSHROOM | SHUTTLE | MAGIC | LETTER | COVERTYPE |
|---|---|---|---|---|---|---|
| INPUT DIMENSION | $2 \times 15$ | $2 \times 23$ | $7 \times 9$ | $2 \times 12$ | $16 \times 26$ | $54 \times 7$ |

We present the experiment results of the classification problem on UCI datasets, `MagicTelescope`, `letter` and `covertype`. The experiment setting and grid search setting for parameter tuning are the same as described in Section 5. For datasets `letter` and `covertype`, only the results of NeuralUCB and NeuralTS with diagonal approximation are reported as the required GPU memory for the calculation of matrix inverse, together with the neural model update, exceeds the supported memory of the NVIDIA GEFORCE RTX 2080 graphical card. Similar observations are obtained for these three datasets. Compared to the linear contextual models, the neural contextual bandit algorithms significantly improve the performance with much lower regret (mistakes on classification). NPR showed consistently better performance to the best neural bandit baselines. And due to its perturbation-based exploration strategy, no extra computational resources are needed, which makes it more practical in real-world problems, especially large-scale tasks.

## B   BACKGROUND FOR THEORETICAL ANALYSIS

We denote the set of all the possible $K$ arms by their context vectors as $\{\mathbf{x}_i\}_{i=1}^{K}$. Our analysis is built upon the existing work in neural tangent kernel techniques.

**Definition B.1** (Jacot et al. (2018); Cao & Gu (2019)). Let $\{\mathbf{x}_i\}_{i=1}^{K}$ be a set of contexts. Define

$$\widetilde{\mathbf{H}}_{i,j}^{(1)} = \mathbf{\Sigma}_{i,j}^{(1)} = \langle \mathbf{x}_i, \mathbf{x}_j \rangle, \qquad \mathbf{B}_{i,j}^{(l)} = \begin{pmatrix} \mathbf{\Sigma}_{i,i}^{(l)} & \mathbf{\Sigma}_{i,j}^{(l)} \\ \mathbf{\Sigma}_{i,j}^{(l)} & \mathbf{\Sigma}_{j,j}^{(l)} \end{pmatrix},$$

$$\mathbf{\Sigma}_{i,j}^{(l+1)} = 2\mathbb{E}_{(u,v)\sim N(\mathbf{0},\mathbf{B}_{i,j}^{(l)})} \left[ \phi(u)\phi(v) \right],$$

$$\widetilde{\mathbf{H}}_{i,j}^{(l+1)} = 2\widetilde{\mathbf{H}}_{i,j}^{(l)}\mathbb{E}_{(u,v)\sim N(\mathbf{0},\mathbf{B}_{i,j}^{(l)})} \left[ \phi'(u)\phi'(v) \right] + \mathbf{\Sigma}_{i,j}^{(l+1)}.$$

Then, $\mathbf{H} = (\widetilde{\mathbf{H}}^{(L)} + \mathbf{\Sigma}^{(L)})/2$ is called the *neural tangent kernel (NTK)* matrix on the context set.

With Definition B.1, we also have the following assumption on the contexts: $\{\mathbf{x}_i\}_{i=1}^{K}$.

**Assumption B.2.** $\mathbf{H} \succeq \lambda_0 \mathbf{I}$; and moreover, for any $1 \leq i \leq K$, $\|\mathbf{x}_i\|_2 = 1$ and $[\mathbf{x}_i]_j = [\mathbf{x}_i]_{j+d/2}$.

By assuming $\mathbf{H} \succeq \lambda_0 \mathbf{I}$, the neural tangent kernel matrix is non-singular (Du et al., 2019; Arora et al., 2019; Cao & Gu, 2019). It can be easily satisfied when *no* two context vectors in the context set are in parallel. The second assumption is just for convenience in analysis and can be easily satisfied by: for any context vector $\mathbf{x}$, $\|\mathbf{x}\|_2 = 1$, we can construct a new context $\mathbf{x}' = [\mathbf{x}^\top, \mathbf{x}^\top]^\top/\sqrt{2}$. It can be verified that if $\boldsymbol{\theta}_0$ is initialized as in Algorithm 1, then $f(\mathbf{x}_i; \boldsymbol{\theta}_0) = 0$ for any $i \in [K]$.

The NTK technique builds a connection between the analysis of DNN and kernel methods. With the following definition of effective dimension, we are able to analyze the neural network by some complexity measures previously used for kernel methods.

**Definition B.3.** The effective dimension $\widetilde{d}$ of the neural tangent kernel matrix on contexts $\{\mathbf{x}_i\}_{i=1}^{K}$ is defined as

$$\widetilde{d} = \frac{\log \det(\mathbf{I} + T\mathbf{H}/\lambda)}{\log(1 + TK/\lambda)}.$$

The notion of effective dimension is first introduced in Valko et al. (2013) for analyzing kernel contextual bandits, which describes the actual underlying dimension in the set of observed contexts $\{\mathbf{x}_i\}_{i=1}^{K}$.

And we also need the following concentration bound on Gaussian distribution.

**Lemma B.4.** Consider a normally distributed random variable $X \sim \mathcal{N}(\mu, \sigma^2)$ and $\beta \geq 0$. The probability that $X$ is within a radius of $\beta\sigma$ from its mean can be written as,

$$\mathbb{P}(|X - \mu| \leq \beta\sigma) \geq 1 - \exp(-\beta^2/2).$$

## C   PROOF OF THEOREMS AND LEMMAS IN SECTION 4

### C.1   PROOF OF LEMMA 4.2

*Proof of Lemma 4.2.* By Lemma 4.1, with a satisfied $m$, with probability at least $1 - \delta$, we have for any $i \in [K]$,

$$h(\mathbf{x}_i) = \langle \mathbf{g}(\mathbf{x}_i; \boldsymbol{\theta}_0)/\sqrt{m}, \sqrt{m}(\boldsymbol{\theta}^* - \boldsymbol{\theta}_0) \rangle, \text{ and } \sqrt{m}\|\boldsymbol{\theta}^* - \boldsymbol{\theta}_0\|_2 \leq \sqrt{2\mathbf{h}^\top \mathbf{H}^{-1}\mathbf{h}} \leq S.$$

Hence, based on the definition of $\mathbf{A}_t$ and $\bar{\mathbf{b}}_t$, we know that $\mathbf{A}_t^{-1}\bar{\mathbf{b}}_t$ is the least square solution on top of the observation noise. Therefore, by Theorem 1 in Abbasi-Yadkori et al. (2011), with probability at least $1 - \delta$, for any $1 \leq t \leq T$, $\boldsymbol{\theta}^*$ satisfies that

$$\|\sqrt{m}(\boldsymbol{\theta}^* - \boldsymbol{\theta}_0) - \mathbf{A}_t^{-1}\bar{\mathbf{b}}_t\|_{\mathbf{A}_t} \leq \sqrt{R^2 \log \frac{\det(\mathbf{A}_t)}{\delta^2 \det(\lambda \mathbf{I})}} + \sqrt{\lambda}S, \tag{C.1}$$

where $R$ is the sub-Gaussian variable for the observation noise $\eta$. Then we have,

$$
\begin{aligned}
&|\langle \mathbf{g}(\mathbf{x}_{t,i}; \boldsymbol{\theta}_0), \mathbf{A}_t^{-1}\bar{\mathbf{b}}_t/\sqrt{m}\rangle - h(\mathbf{x}_{t,i})| \\
=&|\langle \mathbf{g}(\mathbf{x}_{t,i}; \boldsymbol{\theta}_0), \mathbf{A}_t^{-1}\bar{\mathbf{b}}_t/\sqrt{m}\rangle - \langle \mathbf{g}(\mathbf{x}_{t,i}; \boldsymbol{\theta}_0), \boldsymbol{\theta}^* - \boldsymbol{\theta}_0\rangle| \\
\leq&\|\sqrt{m}(\boldsymbol{\theta}^* - \boldsymbol{\theta}_0) - \mathbf{A}_t^{-1}\bar{\mathbf{b}}_t\|_{\mathbf{A}_t}\|\mathbf{g}(\mathbf{x}_{t,i}; \boldsymbol{\theta}_0)/\sqrt{m}\|_{\mathbf{A}_t^{-1}} \leq \alpha_t\|\mathbf{g}(\mathbf{x}_{t,i}; \boldsymbol{\theta}_0)/\sqrt{m}\|_{\mathbf{A}_t^{-1}}.
\end{aligned}
$$

with $\alpha_t = \sqrt{R^2 \log \frac{\det(\mathbf{A}_t)}{\delta^2 \det(\lambda \mathbf{I})}} + \sqrt{\lambda}S$. This completes the proof. $\qquad\square$

## C.2 PROOF OF LEMMA 4.3

The proof starts with three lemmas that bound the error terms of the function value and gradient of a neural network.

**Lemma C.1** (Lemma B.2, Zhou et al. (2020)). There exist constants $\{\bar{C}_i\}_{i=1}^5 > 0$ such that for any $\delta > 0$, with $J$ as the number of steps for gradient descent in neural network learning, if for all $t \in [T]$, $\eta$ and $m$ satisfy

$$
\begin{aligned}
&2\sqrt{t/(m\lambda)} \geq \bar{C}_1 m^{-3/2} L^{-3/2}[\log(KL^2/\delta)]^{3/2}, \\
&2\sqrt{t/(m\lambda)} \leq \bar{C}_2 \min\left\{L^{-6}[\log m]^{-3/2}, \left(m(\lambda\eta)^2 L^{-6} t^{-1}(\log m)^{-1}\right)^{3/8}\right\}, \\
&\eta \leq \bar{C}_3(m\lambda + tmL)^{-1}, \\
&m^{1/6} \geq \bar{C}_4\sqrt{\log m}L^{7/2}t^{7/6}\lambda^{-7/6}(1 + \sqrt{t/\lambda}),
\end{aligned}
$$

then with probability at least $1 - \delta$, we have that $\|\boldsymbol{\theta}_{t-1} - \boldsymbol{\theta}_0\|_2 \leq 2\sqrt{t/(m\lambda)}$ and

$$
\|\boldsymbol{\theta}_{t-1} - \boldsymbol{\theta}_0 - \mathbf{A}_t^{-1}\bar{\mathbf{b}}_t/\sqrt{m}\|_2 \leq (1 - \eta m\lambda)^{J/2}\sqrt{t/(m\lambda)} + \bar{C}_5 m^{-2/3}\sqrt{\log m}L^{7/2}t^{5/3}\lambda^{-5/3}(1 + \sqrt{t/\lambda}).
$$

**Lemma C.2** (Lemma B.3 Zhou et al. (2020)). There exist constants $\{\bar{C}_i^u\}_{i=1}^3 > 0$ such that for any $\delta > 0$, if $\tau$ satisfies

$$
\bar{C}_1^u m^{-3/2} L^{-3/2}[\log(KL^2/\delta)]^{3/2} \leq \tau \leq \bar{C}_2^u L^{-6}[\log m]^{-3/2},
$$

then with probability at least $1 - \delta$, for any $\widetilde{\boldsymbol{\theta}}$ and $\widehat{\boldsymbol{\theta}}$ satisfying $\|\widetilde{\boldsymbol{\theta}} - \boldsymbol{\theta}_0\|_2 \leq \tau, \|\widehat{\boldsymbol{\theta}} - \boldsymbol{\theta}_0\|_2 \leq \tau$ and $j \in [K]$ we have

$$
\left|f(\mathbf{x}_j; \widetilde{\boldsymbol{\theta}}) - f(\mathbf{x}_j; \widehat{\boldsymbol{\theta}}) - \langle \mathbf{g}(\mathbf{x}_j; \widehat{\boldsymbol{\theta}}), \widetilde{\boldsymbol{\theta}} - \widehat{\boldsymbol{\theta}}\rangle\right| \leq \bar{C}_3^u \tau^{4/3} L^3\sqrt{m\log m}.
$$

**Lemma C.3** (Lemma B.4, Zhou et al. (2020)). There exist constants $\{\bar{C}_i^v\}_{i=1}^3 > 0$ such that for any $\delta > 0$, if $\tau$ satisfies that

$$
\bar{C}_1^v m^{-3/2} L^{-3/2}[\log(KL^2/\delta)]^{3/2} \leq \tau \leq \bar{C}_2^v L^{-6}[\log m]^{-3/2},
$$

then with probability at least $1 - \delta$, for any $\|\boldsymbol{\theta} - \boldsymbol{\theta}_0\|_2 \leq \tau$ and $j \in [K]$ we have $\|\mathbf{g}(\mathbf{x}_j; \boldsymbol{\theta})\|_F \leq \bar{C}_3^v\sqrt{mL}$.

With above lemmas, we prove Lemma 4.3 as follows.

*Proof of Lemma 4.3.* According to Lemma C.1, with satisfied neural network and the learning rate $\eta$, we have $\|\boldsymbol{\theta}_{t-1} - \boldsymbol{\theta}_0\|_2 \leq 2\sqrt{t/m\lambda}$. Further, by the choice of $m$, Lemma C.2 and C.3 hold.

Therefore, $|f(\mathbf{x}_{t,i}; \boldsymbol{\theta}_{t-1}) - \langle \mathbf{g}(\mathbf{x}_{t,i}; \boldsymbol{\theta}_0), \mathbf{A}_t^{-1}\bar{\mathbf{b}}_t/\sqrt{m}\rangle|$ can be first upper bounded by:

$$
\begin{aligned}
&|f(\mathbf{x}_{t,i}; \boldsymbol{\theta}_{t-1}) - \langle \mathbf{g}(\mathbf{x}_{t,i}; \boldsymbol{\theta}_0), \mathbf{A}_t^{-1}\bar{\mathbf{b}}_t/\sqrt{m}\rangle| \\
\leq&|f(\mathbf{x}_{t,i}; \boldsymbol{\theta}_{t-1}) - f(\mathbf{x}_{t,i}; \boldsymbol{\theta}_0) - \langle \mathbf{g}(\mathbf{x}_{t,i}; \boldsymbol{\theta}_0), \boldsymbol{\theta}_{t-1} - \boldsymbol{\theta}_0\rangle| + |\langle \mathbf{g}(\mathbf{x}_{t,i}; \boldsymbol{\theta}_0), \boldsymbol{\theta}_{t-1} - \boldsymbol{\theta}_0 - \mathbf{A}_t^{-1}\bar{\mathbf{b}}_t/\sqrt{m}\rangle| \\
\leq&C_{\epsilon,1} m^{-1/6} T^{2/3}\lambda^{-2/3}L^3\sqrt{\log m} + |\langle \mathbf{g}(\mathbf{x}_{t,i}; \boldsymbol{\theta}_0), \boldsymbol{\theta}_{t-1} - \boldsymbol{\theta}_0 - \mathbf{A}_t^{-1}\bar{\mathbf{b}}_t/\sqrt{m}\rangle| \qquad\text{(C.2)}
\end{aligned}
$$

where the first inequality holds due to triangle inequality, $f(\mathbf{x}_{t,i}; \boldsymbol{\theta}_0) = 0$ by the random initialization of $\boldsymbol{\theta}_0$, and the second inequality holds due to Lemma C.2 with the fact $\|\boldsymbol{\theta}_{t-1} - \boldsymbol{\theta}_0\|_2 \leq 2\sqrt{t/m\lambda}$.

Next, we bound the second term of Eq (C.2),

$$|\langle \mathbf{g}(\mathbf{x}_{t,i}; \boldsymbol{\theta}_0), \boldsymbol{\theta}_{t-1} - \boldsymbol{\theta}_0 - \mathbf{A}_t^{-1}\bar{\mathbf{b}}_t/\sqrt{m}\rangle|$$
$$\leq |\langle \mathbf{g}(\mathbf{x}_{t,i}; \boldsymbol{\theta}_0), \boldsymbol{\theta}_{t-1} - \boldsymbol{\theta}_0 - \mathbf{A}_t^{-1}\mathbf{b}_t/\sqrt{m}\rangle| + |\langle \mathbf{g}(\mathbf{x}_{t,i}; \boldsymbol{\theta}_0), \mathbf{A}_t^{-1}\mathbf{b}_t/\sqrt{m} - \mathbf{A}_t^{-1}\bar{\mathbf{b}}_t/\sqrt{m}\rangle|$$
$$\leq \|\mathbf{g}(\mathbf{x}_{t,i}; \boldsymbol{\theta}_0)\|_2 \|\boldsymbol{\theta}_{t-1} - \boldsymbol{\theta}_0 - \mathbf{A}_t^{-1}\mathbf{b}_t/\sqrt{m}\|_2 + |\langle \mathbf{g}(\mathbf{x}_{t,i}; \boldsymbol{\theta}_0), \mathbf{A}_t^{-1}\mathbf{b}_t/\sqrt{m} - \mathbf{A}_t^{-1}\bar{\mathbf{b}}_t/\sqrt{m}\rangle|$$
$$\leq C_{\epsilon,2}(1 - \eta m \lambda)^J \sqrt{TL/\lambda} + C_{\epsilon,3} m^{-1/6} T^{5/3} \lambda^{-5/3} L^4 \sqrt{\log m}(1 + \sqrt{T/\lambda})$$
$$+ |\langle \mathbf{g}(\mathbf{x}_{t,i}; \boldsymbol{\theta}_0), \mathbf{A}_t^{-1}\mathbf{b}_t/\sqrt{m} - \mathbf{A}_t^{-1}\bar{\mathbf{b}}_t/\sqrt{m}\rangle| \qquad (\text{C.3})$$

where the first inequality holds due to the triangle inequality, the second inequality holds according to Cauchy–Schwarz inequality, and the third inequality holds due to Lemma C.1, C.3, with the fact that $\|\boldsymbol{\theta}_{t-1} - \boldsymbol{\theta}_0\|_2 \leq 2\sqrt{t/m\lambda}$ and $t \leq T$.

Now, we provide the upper bound of the last term in Eq (C.3). Based on the definition of $\mathbf{b}_t$ and $\bar{\mathbf{b}}_t$, we have

$$|\langle \mathbf{g}(\mathbf{x}_{t,i}; \boldsymbol{\theta}_0), \mathbf{A}_t^{-1}\mathbf{b}_t/\sqrt{m} - \mathbf{A}_t^{-1}\bar{\mathbf{b}}_t/\sqrt{m}\rangle| = |\langle \mathbf{g}(\mathbf{x}_{t,i}; \boldsymbol{\theta}_0)/\sqrt{m}, \mathbf{A}_t^{-1} \sum_{s=1}^{t-1} \gamma_s^{t-1} \mathbf{g}(\mathbf{x}_{s,a_s}; \boldsymbol{\theta}_0)\rangle|$$

According to the definition of the added random noise $\gamma_s^t \sim \mathcal{N}(0, \sigma^2)$, and Lemma B.4, with $\beta_t \geq 0$, we have the following inequality,

$$\mathbb{P}_t\left(|\langle \mathbf{g}(\mathbf{x}_{t,i}; \boldsymbol{\theta}_0)/\sqrt{m}, \mathbf{A}_t^{-1} \sum_{s=1}^{t-1} \gamma_s^{t-1} \mathbf{g}(\mathbf{x}_{s,a_s}; \boldsymbol{\theta}_0)/\sqrt{m}\rangle| \geq \beta_t \|\mathbf{g}(\mathbf{x}_{t,i}; \boldsymbol{\theta}_0)/\sqrt{m}\|_{\mathbf{A}_t^{-1}}\right)$$
$$\leq 2\exp\left(-\frac{\beta_t^2 \|\mathbf{g}(\mathbf{x}_{t,i}; \boldsymbol{\theta}_0)/\sqrt{m}\|_{\mathbf{A}_t^{-1}}^2}{2\sigma^2 \mathbf{g}(\mathbf{x}_{t,i}; \boldsymbol{\theta}_0)^\top \mathbf{A}_t^{-1}(\sum_{s=1}^{t-1} \mathbf{g}(\mathbf{x}_{s,a_s}; \boldsymbol{\theta}_0)\mathbf{g}(\mathbf{x}_{s,a_s}; \boldsymbol{\theta}_0)^\top/m)\mathbf{A}_t^{-1}\mathbf{g}(\mathbf{x}_{t,i}; \boldsymbol{\theta}_0)/m}\right)$$
$$\leq 2\exp\left(-\frac{\beta_t^2}{2\sigma^2}\right),$$

where the last inequality stands as,

$$\mathbf{g}(\mathbf{x}_{t,i}; \boldsymbol{\theta}_0)^\top \mathbf{A}_t^{-1}(\sum_{s=1}^{t-1} \mathbf{g}(\mathbf{x}_{s,a_s}; \boldsymbol{\theta}_0)\mathbf{g}(\mathbf{x}_{s,a_s}; \boldsymbol{\theta}_0)^\top/m)\mathbf{A}_t^{-1}\mathbf{g}(\mathbf{x}_{t,i}; \boldsymbol{\theta}_0)/m$$
$$\leq \mathbf{g}(\mathbf{x}_{t,i}; \boldsymbol{\theta}_0)^\top \mathbf{A}_t^{-1}(\sum_{s=1}^{t-1} \mathbf{g}(\mathbf{x}_{s,a_s}; \boldsymbol{\theta}_0)\mathbf{g}(\mathbf{x}_{s,a_s}; \boldsymbol{\theta}_0)^\top/m + \lambda\mathbf{I})\mathbf{A}_t^{-1}\mathbf{g}(\mathbf{x}_{t,i}; \boldsymbol{\theta}_0)/m$$
$$= \|\mathbf{g}(\mathbf{x}_{t,i}; \boldsymbol{\theta}_0)/\sqrt{m}\|_{\mathbf{A}_t^{-1}}^2.$$

Therefore, in round $t$, for the given arm $i$,

$$\mathbb{P}_t\left(|\langle \mathbf{g}(\mathbf{x}_{t,i}; \boldsymbol{\theta}_0), \mathbf{A}_t^{-1}\mathbf{b}_t/\sqrt{m} - \mathbf{A}_t^{-1}\bar{\mathbf{b}}_t/\sqrt{m}\rangle| \leq \beta_t \|\mathbf{g}(\mathbf{x}_{t,i}; \boldsymbol{\theta}_0)/\sqrt{m}\|_{\mathbf{A}_t^{-1}}\right) \geq 1 - \exp(-\beta_t^2/(2\sigma^2)).$$

Taking the union bound over $K$ arms, we have that for any $t$:

$$\mathbb{P}_t\left(\forall i \in [K], |\langle \mathbf{g}(\mathbf{x}_{t,i}; \boldsymbol{\theta}_0), \mathbf{A}_t^{-1}\mathbf{b}_t/\sqrt{m} - \mathbf{A}_t^{-1}\bar{\mathbf{b}}_t/\sqrt{m}\rangle| \leq \beta_t \|\mathbf{g}(\mathbf{x}_{t,i}; \boldsymbol{\theta}_0)/\sqrt{m}\|_{\mathbf{A}_t^{-1}}\right) \geq 1 - K\exp(-\beta_t^2/(2\sigma^2)).$$

By choosing $\beta_t = \sigma\sqrt{4\log t + 2\log K}$, we have:

$$\mathbb{P}_t\left(\forall i \in [K], |\langle \mathbf{g}(\mathbf{x}_{t,i}; \boldsymbol{\theta}_0), \mathbf{A}_t^{-1}\mathbf{b}_t/\sqrt{m} - \mathbf{A}_t^{-1}\bar{\mathbf{b}}_t/\sqrt{m}\rangle| \leq \beta_t \|\mathbf{g}(\mathbf{x}_{t,i}; \boldsymbol{\theta}_0)/\sqrt{m}\|_{\mathbf{A}_t^{-1}}\right) \geq 1 - \frac{1}{t^2}.$$

This completes the proof. $\qquad \square$

### C.3 PROOF OF LEMMA 4.4

The proof requires the anti-concentration bound of Gaussian distribution.

**Lemma C.4.** For a Gaussian random variable $X \sim N(\mu, \sigma^2)$, for any $\beta > 0$,

$$\mathbb{P}\left(\frac{X - \mu}{\sigma} > \beta\right) \geq \frac{\exp(-\beta^2)}{4\sqrt{\pi}\beta}.$$

*Proof of Lemma 4.4.* Under event $E_{t,2}$, according to Lemma 4.2, (C.2) and (C.3), we have, for all $i \in [K]$,

$$\mathbb{P}_t(f(\mathbf{x}_{t,i}; \boldsymbol{\theta}_t) > h(\mathbf{x}_{t,a_t^*}) - \epsilon(m))$$
$$\geq \mathbb{P}_t(\langle \mathbf{g}(\mathbf{x}_{t,i}; \boldsymbol{\theta}_0), \mathbf{A}_t^{-1} \mathbf{b}_t / \sqrt{m} \rangle > \langle \mathbf{g}(\mathbf{x}_{t,i}; \boldsymbol{\theta}_0), \mathbf{A}_t^{-1} \bar{\mathbf{b}}_t / \sqrt{m} \rangle + \alpha_t \|\mathbf{g}(\mathbf{x}_{t,i}; \boldsymbol{\theta}_0) / \sqrt{m}\|_{\mathbf{A}_t^{-1}})$$

According to the definition of $\mathbf{A}_t$, $\bar{\mathbf{b}}_t$ and $\mathbf{b}_t$, we have,

$$\langle \mathbf{g}(\mathbf{x}_{t,i}; \boldsymbol{\theta}_0), \mathbf{A}_t^{-1} \mathbf{b}_t / \sqrt{m} - \mathbf{A}_t^{-1} \bar{\mathbf{b}}_t / \sqrt{m} \rangle = \sum_{s=1}^t \gamma_s^t \cdot \frac{1}{m} \mathbf{g}(\mathbf{x}_{t,i}; \boldsymbol{\theta}_0)^\top \mathbf{A}_t^{-1} \mathbf{g}(\mathbf{x}_{s,a_s}; \boldsymbol{\theta}_0) = U_t$$

which follows a Gaussian distribution with mean $\mathbb{E}[U_t] = 0$ and variance $\text{Var}[U_t]$ as:

$$\text{Var}[U_t] = \sigma^2 \cdot \left( \sum_{s=1}^t \left( \frac{1}{m} \mathbf{g}(\mathbf{x}_{t,i}; \boldsymbol{\theta}_0)^\top \mathbf{A}_t^{-1} \mathbf{g}(\mathbf{x}_{s,a_s}; \boldsymbol{\theta}_0) \right)^2 \right)$$
$$= \sigma^2 \|\mathbf{g}(\mathbf{x}_{t,i}; \boldsymbol{\theta}_0) / \sqrt{m}\|_{\mathbf{A}_t^{-1}}^2 - \lambda \sigma^2 \|\mathbf{g}(\mathbf{x}_{t,i}; \boldsymbol{\theta}_0) / \sqrt{m}\|_{\mathbf{A}_t^{-2}}^2$$
$$\geq \sigma^2 (1 - \lambda \lambda_{\min}^{-1}(\mathbf{A}_t)) \|\mathbf{g}(\mathbf{x}_{t,i}; \boldsymbol{\theta}_0) / \sqrt{m}\|_{\mathbf{A}_t^{-1}}^2$$
$$\geq \sigma^2 (1 - \lambda \lambda_K^{-1}(\mathbf{A}_K)) \|\mathbf{g}(\mathbf{x}_{t,i}; \boldsymbol{\theta}_0) / \sqrt{m}\|_{\mathbf{A}_t^{-1}}^2,$$

where the second equality holds according to the definition of $\mathbf{A}_t$, and the first inequality holds as for any positive semi-definite matrix $\mathbf{M} \in \mathbb{R}^{d \times d}$,

$$\mathbf{x}^\top \mathbf{M}^2 \mathbf{x} = \lambda_{\max}^2(\mathbf{M}) \mathbf{x}^\top (\lambda_{\max}^{-2}(\mathbf{M}) \mathbf{M}^2) \mathbf{x} \leq \lambda_{\max}(\mathbf{M}) \|\mathbf{x}\|_{\mathbf{M}}^2,$$

and $\lambda_{\min}(\mathbf{A}_t) \geq \lambda_K(\mathbf{A}_K)$. Therefore, based on Lemma C.4, by choosing $\sigma = \alpha_t / \sqrt{1 - \lambda \lambda_K^{-1}(\mathbf{A}_K)}$, the target probability could be lower bounded by,

$$\mathbb{P}_t(f(\mathbf{x}_{t,j}; \boldsymbol{\theta}_t) > h(\mathbf{x}_{t,a_t^*}) - \epsilon(m)) \geq \mathbb{P}_t(U_t > \alpha_t \|\mathbf{g}(\mathbf{x}_{t,i}; \boldsymbol{\theta}_0) / \sqrt{m}\|_{\mathbf{A}_t^{-1}})$$
$$= \mathbb{P}_t \left( \frac{U_t}{\text{Var}[U_t]} > \frac{\alpha_t \|\mathbf{g}(\mathbf{x}_{t,i}; \boldsymbol{\theta}_0) / \sqrt{m}\|_{\mathbf{A}_t^{-1}}}{\text{Var}[U_t]} \right) \geq \mathbb{P}_t \left( \frac{U_t}{\text{Var}[U_t]} > 1 \right) \geq \frac{1}{4e\sqrt{\pi}}$$

This completes the proof. $\square$

## C.4 PROOF OF LEMMA 4.5

*Proof of Lemma 4.5.* With the defined sufficiently sampled arms in round $t$, $\Omega_t$, we have the set of undersampled arms $\bar{\Omega}_t = [K] \setminus \Omega_t$. We also have the least uncertain and undersampled arm $e_t$ in round $t$ defined as,

$$e_t = \underset{j \in \bar{\Omega}_t}{\arg\min} \|\mathbf{g}(\mathbf{x}_{j,t}; \boldsymbol{\theta}_0) / \sqrt{m}\|_{\mathbf{A}_t^{-1}}$$

In round $t$, it is easy to verify that $\mathbb{E}[h(\mathbf{x}_{t,a_t^*}) - h(\mathbf{x}_{t,a_t})] \leq \mathbb{E}[(h(\mathbf{x}_{t,a_t^*}) - h(\mathbf{x}_{t,a_t})) \mathbb{1}\{E_{t,2}\}] + \mathbb{P}_t(\bar{E}_{t,2})$.

Under event $E_{t,1}$ and $E_{t,2}$, we have,

$$h(\mathbf{x}_{t,a_t^*}) - h(\mathbf{x}_{t,a_t})$$
$$= h(\mathbf{x}_{t,a_t^*}) - h(\mathbf{x}_{t,e_t}) + h(\mathbf{x}_{t,e_t}) - h(\mathbf{x}_{t,a_t})$$
$$\leq \Delta_{e_t} + f(\mathbf{x}_{t,e_t}, \boldsymbol{\theta}_t) - f(\mathbf{x}_{t,a_t}, \boldsymbol{\theta}_t) + 2\epsilon(m) + (\beta_t + \alpha_t)(\|\mathbf{g}(\mathbf{x}_{t,e_t}; \boldsymbol{\theta}_0) / \sqrt{m}\|_{\mathbf{A}_t^{-1}} + \|\mathbf{g}(\mathbf{x}_{t,a_t}; \boldsymbol{\theta}_0) / \sqrt{m}\|_{\mathbf{A}_t^{-1}})$$
$$\leq 4\epsilon(m) + (\beta_t + \alpha_t)(2\|\mathbf{g}(\mathbf{x}_{t,e_t}; \boldsymbol{\theta}_0) / \sqrt{m}\|_{\mathbf{A}_t^{-1}} + \|\mathbf{g}(\mathbf{x}_{t,a_t}; \boldsymbol{\theta}_0) / \sqrt{m}\|_{\mathbf{A}_t^{-1}})$$

Next, we will bound $\mathbb{E}[\|\mathbf{g}(\mathbf{x}_{t,e_t}; \boldsymbol{\theta}_0) / \sqrt{m}\|_{\mathbf{A}_t^{-1}}]$. It is easy to see that,

$$\mathbb{E}[\|\mathbf{g}(\mathbf{x}_{t,a_t}; \boldsymbol{\theta}_0) / \sqrt{m}\|_{\mathbf{A}_t^{-1}}]$$
$$= \mathbb{E}[\|\mathbf{g}(\mathbf{x}_{t,a_t}; \boldsymbol{\theta}_0) / \sqrt{m}\|_{\mathbf{A}_t^{-1}} \mathbb{1}\{a_t \in \bar{\Omega}_t\}] + \mathbb{E}[\|\mathbf{g}(\mathbf{x}_{t,a_t}; \boldsymbol{\theta}_0) / \sqrt{m}\|_{\mathbf{A}_t^{-1}} \mathbb{1}\{a_t \in \Omega_t\}]$$
$$\geq \|\mathbf{g}(\mathbf{x}_{t,e_t}; \boldsymbol{\theta}_0) / \sqrt{m}\|_{\mathbf{A}_t^{-1}} \mathbb{P}_t(a_t \in \bar{\Omega}_t).$$

It can be arranged as $\|\mathbf{g}(\mathbf{x}_{t,e_t}; \boldsymbol{\theta}_0)/\sqrt{m}\|_{\mathbf{A}_t^{-1}} \leq \mathbb{E}[\|\mathbf{g}(\mathbf{x}_{t,a_t}; \boldsymbol{\theta}_0)/\sqrt{m}\|_{\mathbf{A}_t^{-1}}]/\mathbb{P}_t(a_t \in \bar{\Omega}_t)$. Hence, the one step regret can be bounded by the following inequality with probability at least $1 - \delta$,

$$\mathbb{E}[h(\mathbf{x}_{t,a_t^*}) - h(\mathbf{x}_{t,a_t})]$$
$$\leq \mathbb{P}(\bar{E}_{t,2}) + 4\epsilon(m) + (\beta_t + \alpha_t)\big(1 + 2/\mathbb{P}(a_t \in \bar{\Omega}_t)\big)\|\mathbf{g}(\mathbf{x}_{t,a_t}; \boldsymbol{\theta}_0)/\sqrt{m}\|_{\mathbf{A}_t^{-1}}.$$

For the probability $\mathbb{P}_t(a_t \in \bar{\Omega}_t)$, we have:

$$\mathbb{P}_t(a_t \in \bar{\Omega}_t) \geq \mathbb{P}_t(\exists a_i \in \bar{\Omega}_t : f(\mathbf{x}_{t,a_i}; \boldsymbol{\theta}_t) > \max_{a_j \in \Omega_t} f(\mathbf{x}_{t,a_j}; \boldsymbol{\theta}_t))$$

$$\geq \mathbb{P}_t(f(\mathbf{x}_{t,a_t^*}; \boldsymbol{\theta}_t) > \max_{a_j \in \Omega_t} f(\mathbf{x}_{t,a_j}; \boldsymbol{\theta}_t))$$

$$\geq \mathbb{P}_t(f(\mathbf{x}_{t,a_t^*}; \boldsymbol{\theta}_t) > \max_{a_j \in \Omega_t} f(\mathbf{x}_{t,a_j}; \boldsymbol{\theta}_t), E_{t,2} \text{ occurs})$$

$$\geq \mathbb{P}_t(f(\mathbf{x}_{t,a_t^*}; \boldsymbol{\theta}_t) > h(\mathbf{x}_{t,a_t^*}) - \epsilon(m)) - \mathbb{P}_t(\bar{E}_{t,2})$$

$$\geq \mathbb{P}_t(E_{t,3}) - \mathbb{P}_t(\bar{E}_{t,2})$$

where the fourth inequality holds as for arm $a_j \in \Omega_t$, under event $E_{t,1}$ and $E_{t,2}$:

$$f(\mathbf{x}_{t,j}; \boldsymbol{\theta}_t) \leq h(\mathbf{x}_{t,j}) + \epsilon(m) + (\beta_t + \alpha_t)\|\mathbf{g}(\mathbf{x}_{t,j}; \boldsymbol{\theta}_0)/\sqrt{m}\|_{\mathbf{A}_t^{-1}} \leq h(\mathbf{x}_{t,a_t^*}) - \epsilon(m).$$

This completes the proof. $\qquad\square$

## C.5 PROOF OF LEMMA 4.6

We first need the following lemma from Abbasi-Yadkori et al. (2011).

**Lemma C.5** (Lemma 11, Abbasi-Yadkori et al. (2011)). *We have the following inequality:*

$$\sum_{t=1}^T \min\left\{\|\mathbf{g}(\mathbf{x}_{t,a_t}; \boldsymbol{\theta}_0)/\sqrt{m}\|_{\mathbf{A}_{t-1}^{-1}}^2, 1\right\} \leq 2\log\frac{\det \mathbf{A}_T}{\det \lambda \mathbf{I}}.$$

**Lemma C.6** (Lemma B.1, Zhou et al. (2020)). *Let $\mathbf{G} = [\mathbf{g}(\mathbf{x}^1; \boldsymbol{\theta}_0), \ldots, \mathbf{g}(\mathbf{x}^K; \boldsymbol{\theta}_0)]/\sqrt{m} \in \mathbb{R}^{p \times K}$. Let $\mathbf{H}$ be the NTK matrix as defined in Definition B.1. For any $\delta \in (0, 1)$, if*

$$m = \Omega\left(\frac{L^6 \log(K^2 L/\delta)}{\epsilon^4}\right),$$

*then with probability at least $1 - \delta$, we have*

$$\|\mathbf{G}^\top \mathbf{G} - \mathbf{H}\|_F \leq K\epsilon.$$

*Proof of Lemma 4.6.* Denote $\mathbf{G} = [\mathbf{g}(\mathbf{x}^1; \boldsymbol{\theta}_0)/\sqrt{m}, \ldots, \mathbf{g}(\mathbf{x}^K; \boldsymbol{\theta}_0)/\sqrt{m}] \in \mathbb{R}^{p \times K}$, then we have

$$\log\frac{\det \mathbf{A}_T}{\det \lambda \mathbf{I}} = \log\det\left(\mathbf{I} + \sum_{t=1}^T \mathbf{g}(\mathbf{x}_{t,a_t}; \boldsymbol{\theta}_0)\mathbf{g}(\mathbf{x}_{t,a_t}; \boldsymbol{\theta}_0)^\top/(m\lambda)\right)$$

$$\leq \log\det\left(\mathbf{I} + \sum_{t=1}^T \sum_{i=1}^K \mathbf{g}(\mathbf{x}^i; \boldsymbol{\theta}_0)\mathbf{g}(\mathbf{x}^i; \boldsymbol{\theta}_0)^\top/(m\lambda)\right)$$

$$= \log\det\left(\mathbf{I} + T\mathbf{G}\mathbf{G}^\top/\lambda\right) = \log\det\left(\mathbf{I} + T\mathbf{G}^\top\mathbf{G}/\lambda\right), \qquad (\text{C.4})$$

where the inequality holds naively, the third equality holds since for any matrix $\mathbf{A} \in \mathbb{R}^{p \times K}$, we have $\det(\mathbf{I} + \mathbf{A}\mathbf{A}^\top) = \det(\mathbf{I} + \mathbf{A}^\top\mathbf{A})$. We can further bound Eq (C.4) as follows:

$$\log\det\left(\mathbf{I} + T\mathbf{G}^\top\mathbf{G}/\lambda\right) = \log\det\left(\mathbf{I} + T\mathbf{H}/\lambda + T(\mathbf{G}^\top\mathbf{G} - \mathbf{H})/\lambda\right)$$

$$\leq \log\det\left(\mathbf{I} + T\mathbf{H}/\lambda\right) + \langle(\mathbf{I} + T\mathbf{H}/\lambda)^{-1}, T(\mathbf{G}^\top\mathbf{G} - \mathbf{H})/\lambda\rangle$$

$$\leq \log\det\left(\mathbf{I} + T\mathbf{H}/\lambda\right) + \|(\mathbf{I} + T\mathbf{H}/\lambda)^{-1}\|_F\|\mathbf{G}^\top\mathbf{G} - \mathbf{H}\|_F \cdot T/\lambda$$

$$\leq \log\det\left(\mathbf{I} + T\mathbf{H}/\lambda\right) + T\sqrt{K}\|\mathbf{G}^\top\mathbf{G} - \mathbf{H}\|_F$$

$$\leq \log\det\left(\mathbf{I} + T\mathbf{H}/\lambda\right) + 1 = \widetilde{d}\log(1 + TK/\lambda) + 1,$$

where the first inequality holds due to the concavity of $\log \det(\cdot)$, the second inequality holds due to the fact that $\langle \mathbf{A}, \mathbf{B} \rangle \leq \|\mathbf{A}\|_F \|\mathbf{B}\|_F$, the third inequality holds due to the facts that $\mathbf{I} + \mathbf{H}/\lambda \succeq \mathbf{I}$, $\lambda \geq 1$ and $\|\mathbf{A}\|_F \leq \sqrt{K}\|\mathbf{A}\|_2$ for any $\mathbf{A} \in \mathbb{R}^{K \times K}$, the fourth inequality holds by Lemma C.6 with the required choice of $m$, the fifth inequality holds by the definition of effective dimension in Definition B.3. This completes the proof. $\qquad\square$

