# OpenReview forum: "Learning Neural Contextual Bandits through Perturbed Rewards"
_ICLR.cc/2022/Conference — ICLR 2022 Poster_

### Official Review · Reviewer_ha3h · 2021-10-28

**Correctness:** 3
**Technical Novelty And Significance:** 3
**Empirical Novelty And Significance:** 3
**Recommendation:** 8
**Confidence:** 3

**Main Review:**

PROS:

•	The paper is very nicely written. I found it very easy to follow even the more complex aspects and the introduction does a good job of motivating the work and providing a useful history of the relevant literature.

•	The central idea of the paper is a useful and sensible combination and extension of existing techniques. It seems to work effectively, in addition to having desirable theoretical properties, and should be interesting to the ICLR community.

MAJOR COMMENTS ON CONS:

•	What can be said in terms of order results on the computational cost of NPR versus NeuralUCB/NeuralTS? The empirical comparison is useful, but to have both would solidify the core argument for the usefulness of NPR and be quite informative.

•	The biggest issue I can see is that the experimental results are not as informative as they could be. Firstly, they do not give a sense of the distribution of regret over different runs, and second, they do not (in the fully synthetic scenario) seem to be run for long enough to show that the algorithms have converged. All the regret plots look linear but with differing gradients in fig 1. Can you remedy this?

•	The use of grid-search to select $\eta$ perhaps needs some commentary as the theoretical guarantees section talks about a specific value $\eta$ for the theoretical guarantees to hold. The diagonalised algorithms are criticised for losing theoretical guarantees, but does this optimisation of $\eta$ not affect the theoretical guarantees of your approach?

MINOR COMMENTS ON CONS

•	The definition of $f(x,\theta)$ may be better in an equation display as the page break in the middle of it is unfortunate.

•	Middle of page 3: should be “When all the $K$ arms have been pulled once” not “for once”.

•	If I were aiming to improve the commentary on the theory, I would perhaps add a paragraph to the start of Section 4 to highlight what the main challenges/novelties in the theory are – as I understand this it is the design of the noise and of the step-size in the gradient descent? This would help the reader get a clear picture of what is more/less standard going in to this section.

•	The regret bound achieved in Theorem 4.7 is (understandably) quite complex and not directly compared to the regret of the competitor algorithms. I think there would be value in giving some commentary afterwards to highlight the dependence on the main terms such as T and K and \tilde{d} and how this compares to NeuralUCB/TS.


**Summary Of The Paper:**

The paper considers a version of the stochastic contextual bandit where the underlying reward function is modelled by a deep neural network. This framework is useful for a number of complex problems where simpler parametric models of the reward are insufficient. The authors propose an algorithm inspired by the perturbed reward approaches of Kveton et al (2019a,b,2020) which ensures an exploration-exploitation balance by adding additional noise to the observed rewards. This approach avoids the inversion of large covariance matrices which existing approaches (NeuralUCB and NeuralTS) necessitate, and results in a computational speed-up while maintaining near-optimal regret guarantees.

**Summary Of The Review:**

I’m very positive about the idea behind this paper: to use combine perturbed rewards with neural contextual bandits and think that this work is timely and potentially very impactful. I have a few reservations about the extent to which experiments support the main claims of the paper, and as such have chosen a borderline score which I could increase if my concerns are addressed.

---

> ### Author Response · Authors · 2021-11-17
> **Response to Reviewer ha3h**
>
> We sincerely thank the reviewer for the comments and suggestions. Please find our response in the following.
>
> **What can be said in terms of order results on the computational cost of NPR versus NeuralUCB/NeuralTS? The empirical comparison is useful, but to have both would solidify the core argument for the usefulness of NPR and be quite informative.**
>
> In our analysis, p represents the number of parameters in the neural network. Therefore, in each round, a neural bandit algorithm generally does the following: (1) predict reward of each arm based on the current neural network, (2) select the arm according to the algorithm’s exploration strategy, (3) update the model according to the feedback.
> For all the neural bandit algorithms, step (1) and step (3) have the time complexity of \\(O(Kp)\\) and \\(O(nKP)\\), where \\(n\\) is the number of epochs in the model update process, \\(K\\) is the number of candidate arms, and \\(p\\) is the number of parameters in neural network. In step (2), for NeuralUCB and NeuralTS, the inverse of the covariance matrix is required, which has the time complexity of \\(O(p^3)\\). Besides, constructing the confidence interval has the time complexity of \\(O(Kp^2)\\) for each arm in NeuralUCB and NeuralTS. While for NPR, no extra computation is needed besides predicting the reward in step (2).
> Therefore, the total time complexity of NeuralUCB/NeuralTS is \\(O(Kp+nKp+p^3+Kp^2+K) = O(p^3)\\), while the total time complexity of NPR is \\(O(Kp + nKp + K) = O(nKp)\\), as p nominates all other constants in this complexity analysis.
>
> **The biggest issue I can see is that the experimental results are not as informative as they could be. Firstly, they do not give a sense of the distribution of regret over different runs, and second, they do not (in the fully synthetic scenario) seem to be run for long enough to show that the algorithms have converged. All the regret plots look linear but with differing gradients in fig 1. Can you remedy this?**
>
> The purpose of our experiments on synthetic data is to compare both the computational efficiency and effectiveness of NPR with the baselines, because our neural bandit baselines cannot afford a large NN model due to their huge time complexity. Therefore, for the neural models, we chose a relatively small network width, e.g., \\(m = 64\\), to make sure that NeuralUCB/NeuralTS can be run with a full covariance matrix (described in the original paper) in our initial experiments. The results showed that all the neural models achieved better performance than their linear counterparts. However, such a small neural network unfortunately failed to capture the non-linear relationship, as shown in the original paper (and pointed out by the reviewer). We now re-run the experiments with a more complicated neural network, e.g., set neural network width \\(m = 128\\) for each hidden layer. But due to the limited GPU memory, it is now impossible to run NeuralUCB and NeuralTS with full covariance matrix. We compared NPR and the other baselines in the newly added figures (in Section A.1.1 of the appendix of our updated paper), and we can observe NPR showed much better performance than the NeuralUCB/NeuralTS, especially the sublinear regret in NPR.
>
> To ensure visibility of our result figures, we decided not to include variances. We plot the variance of the newly updated result, in Figure 4 in the appendix. We can observe the shaded areas for variance seriously overlap with each other and affect the readability of the results. Hence, we still prefer not to include them in our figures.
>
> **The regret bound achieved in Theorem 4.7 is (understandably) quite complex and not directly compared to the regret of the competitor algorithms.**
>
> The upper regret bound of NPR is in the order of \\(\tilde{O}(\tilde{d}\sqrt{T})\\), which is in the same order as NeuralUCB and NeuralTS, in terms of those important dominating terms. But as explained in the paper and our responses, NPR’s main advantage is its much lighter computational complexity.
>
> **What the main challenges/novelties in the theory are?**
>
> As we explained in the paper and our responses before, the main technical challenge is to precisely compute the minimally needed noise level that enables sufficient exploration and introduces least impact in the regret. And to achieve so, we have to carefully decompose the deviations caused by different sources in the estimated reward, e.g., the observation noise, the added pseudo noise, and the neural network approximation. These constitute our main technical contribution.

---

> > ### Comment · Reviewer_ha3h · 2021-11-20
> > **Post author Response**
> >
> > Thanks for these clarifications and in particular adding the more extensive experiments. this addresses my main concern, and in light of this and the other reviewers I will join reviewer NMyc in raising my score.
> >
> > I think the paper is interesting and well written and commend the authors for this.

---

> > > ### Author Response · Authors · 2021-11-23
> > > **Response to Reviewer ha3h**
> > >
> > > Thank you for your helpful suggestions and encouragement!

---

### Official Review · Reviewer_NMyc · 2021-11-01

**Correctness:** 4
**Technical Novelty And Significance:** 2
**Empirical Novelty And Significance:** 2
**Recommendation:** 8
**Confidence:** 3

**Main Review:**

__Pros.__

1. The algorithm is practical as the neural network training is unaffected by the exploration strategy. This is a big advantage over NeuralTS and NeuralUCB.

2. The paper uses the same idea as [Kveton et al 2020] however it rigorously proves regret guarantees for the neural network function class under the NTK setting.

3. The regret guarantee only scales with the effective dimension (Valko et al 2013) of the NTK.

4. The experiments show some promise in real world datasets especially because of the predictably better running times than NeuralUCB.

__Cons.__

I have some general clarifying questions and some comments, which will hopefully improve the quality of the paper.

1. My biggest concern is that the paper fails to discus and compare with the line of work in FALCON (https://arxiv.org/pdf/2003.12699.pdf). Both the papers lie in the realizable setting. FALCON also does not need to modify the training of the neural network in any way and in fact needs to train the model even less frequently based on an epoch schedule. Further I think the guarantees of FALCON will probably hold with a similar regret term; basically the generalization error from the NTK will appear in Assumption 2 and Theorem 2 in https://arxiv.org/pdf/2003.12699.pdf. I think the paper should definitely have a discussion with this. Besides the paper should also compare with this algorithm in the experiments as it is easy to implement.

2. The $\lambda$ term is missing from Equation 3.2. In algorithm 1 how is $\nu$ set -- can the authors please point me to wherever this is defined in the paper? What setting of $\nu$ is used in practice in the experiments? Also, in practice is the neural network trained per time-step as in the algorithm pseudocode? That would be pretty prohibitive in practice.

3. (minor) In the def of $\bar{b}_t$ in page 4 the expectation of $\gamma_s^{t-1}$ is zero, right?  Then is this just for the sake of readability?

4. The reference to definition of effective dimension is broken in page 5.

5. In the synthetic experiments in Figure 1 it does seem that all algorithms kind of have a linear regret scaling. It would be great if we can add a full information algorithm (algorithm that sees reward from all arms chooses greedily). Such an oracle algorithm would show the lower bound on the achievable regret. I could find that the real experiments in Figure 2 are averaged over 10 runs, is such an averaging performed in the simulated setting. If so, then please also plot the confidence bars.

Also subsampling k arms out of a total of 100 arms each time is slightly non-standard. It would be good if the algorithms are just compared over a fixed set of 10, 20, 50 and 100 arms, that are decided in the beginning and held fixed over the rounds.

6. In all experiments, as before it would be great if the following baselines (or moonshots) can be added
-- FALCON
-- Greedy (bandit feedback plus choosing the arms greedily)
-- Full information (full information feedback and the arms are chosen greedily)


**Summary Of The Paper:**

The paper studies neural contextual bandits in the realizable setting. It proposes an algorithm that trains the neural network that maps arm contexts to rewards, with perturbed rewards. The paper proves that under NTK analysis the learnt function provides an optimistic estimate of the reward of each arms and therefore sub-linear regret guarantees can be derived. Crucially the dimension term in the regret only depends on the effective dimension of the NTK. Some experiments are performed on real and synthetic datasets where the proposed algorithm performs at par with neuralTS and neuralUCB algorithms while cutting down the running time by a large factor. Thus this can indicate the practicality of the proposed algorithm.

**Summary Of The Review:**

I think the pros above outweigh the cons slightly in my mind. I am happy to raise the score if the authors can answer 1 and 6 among the cons.

---

> ### Author Response · Authors · 2021-11-17
> **Response to Reviewer NMyc**
>
> We sincerely thank the reviewer for the comments and suggestions. Please find our response in the following.
>
> 1. Thanks for suggesting this work! FALCON and NPR differ from the following important aspects, though they both share the realizable assumption. First, NPR has a very different problem setup against FALCON. In FALCON, in each round, the context vector \\(x_t\\) is shared by all K arms; but in ours, every arm has its own context vector, which governs the corresponding observed reward. See our problem definition in Section 3 for more details. Second, the dominating term of FALCON’s derived regret has a polynomial dependence on the number of arms K, while our regret only depends on it logarithmically.
> Moreover, the efficiency advantage of FALCON comes from its scheduled update, which is applicable to NPR as well. And as discussed in our response to Reviewer ZmVR, we can further trade computational efficiency with regret via a scheduled update in NPR.
> In addition, we also would like to clarify that neither FALCON+ (its general offline regression oracle version) nor NPR really needs to consider the NTK approximation error, as NTK is only used in analyzing NPR (not even needed in analyzing FALCON+), not to design NPR.  Motivated by the suggestion, we are also empirically comparing NPR with FALCON+. We exactly followed FALCON+’s description with the same NN architecture as in NPR, but found FALCON+’s performance was very poor and unstable. The main reason is that its learning rate \\(\gamma_m\\) depends on the error rate of offline regression oracle, which is the MLP in our experiment. But we only have big-O analysis about its error rate (as suggested in FALCON+’s paper Example 3). Hence hyper-parameter tuning for \\(\gamma_m\\) is frustratingly difficult as the tuning of \\(\gamma_m\\) has to be performed for each epoch. But NPR does not have this issue at all. Currently FALCON+’s performance is only slightly better than a random policy, and much worse than all the compared baselines (even the linear models). We will report the results on this comparison as soon as possible, if we could find any reasonable performance of FALCON+.
>
> 2. (1) First, thanks for pointing out the typo, and we have updated the corresponding part in the revision paper.  (2) The setting of the variance \\(\nu\\) is introduced in Lemma 4.4, which is carefully analyzed to guarantee sufficient deviation in the estimated reward for exploration. As described in Lemma 4.4, the value of \\(\nu\\) depends on \\(\delta\\) in Lemma 4.2, which describes the probability of the concentration behavior of the observation noise. In our experiment, we used grid search in the hyper-parameter \\(\delta\\) for the computed \\(\nu\\). (3) Yes, in our experiments, the model is updated every time-step. This model update is required by the desired convergence of the neural model update to obtain sublinear regret, **which is necessary for all existing neural bandit models with theoretical guarantees**. And hence, NPR does not introduce any added complexity here.
> With incremental model update, the catastrophic forgetting needs to be seriously handled; but the corresponding theoretical analysis is still lacking currently. One possible practical option is to reduce the frequency in the retraining of NN via a scheduled model update. But it is known to be at the cost of increased regret. How to further reduce computational cost in NPR while maintaining its regret is an important future work of ours.
> 3. Yes, the expectation of \\(\gamma_s^{t-1}\\) is 0, and we present it with the expectation for better readability.
> 4. Thanks for pointing it out and we have updated it in the revision.

---

> > ### Comment · Reviewer_NMyc · 2021-11-18
> > **Thanks for the response and doing further experiments.**
> >
> > 1. It is possible to map the general setting in Falcon, SquareCB to a linear bandits kind of setting (your setting with linear function class) using section 2.3 in https://arxiv.org/pdf/2002.04926.pdf (Eq. 9). But because of the generality you do sacrifice on the regret here and get a Poly(K) term in the regret even in the linear bandits setting which is avoidable by linUCB for example. But in my experience and even the experiments in Table 1 of https://arxiv.org/pdf/2010.03104.pdf I have seen IGW based algorithms work well. But actually I have no intuition about what would happen in practice with neural network function classes so I believe the author's experiences. Yes tuning $\gamma_t$ can be trouble some but usually if you set $\gamma_t = C \sqrt{t}$ and then tune $C$ on a small held out section of the dataset, it should work. It would still be great to have that as a baseline even if it performs poorly in these settings as that would be a important datapoint for future research.
> >
> > 2. Is $\nu$ a hyper-parameter in practice that is tuned?
> >
> > 3 and 4. Thanks for the clarification.
> >
> > 5. Ok this new setting makes things more clear.
> >
> > I have decided to raise my score to 8.

---

> > > ### Author Response · Authors · 2021-11-23
> > > **Response to Reviewer NMyc**
> > >
> > > **Yes tuning \\(\gamma_t\\) can be trouble some but usually if you set \\(\gamma_t = C\sqrt{t}\\) and then tune \\(C\\) on a small held out section of the dataset, it should work...**
> > >
> > > Thanks for the great suggestion! We followed the suggestion by setting \\(\gamma_t=C\sqrt{t}\\) for FALCON+, and it indeed led to much better performance than our previous attempts. In particular, we tuned C over \\(\\{50*i\\}_{i=1}^{20}\\) in the first 1000 rounds. Following the same experiment setting in the main paper, the best set of hyper-paramters were then applied to the full simulation over 10000 rounds (i.e., only the chosen trial will be continued for the rest 9000 rounds). Due to the time limit, we only studied FALCON+ on the newly updated experiment settings (i.e., setting m=128 in our synthetic data as in **Figure 4 and Figure 5 in the appendix**). We are working on the experiments on the other datasets, e.g., the classification dataset and the LastFM&Delicious dataset, and will add all the results in our final version.
> > >
> > > In the newly obtained results, we can observe that FALCON+ showed better performance in the initial rounds; but with more iterations, its regret was higher than other neural bandit baselines’ and our NPR model’s. More importantly, compared to other neural bandit models, FALCON+ had a much slower convergence in regret, e.g., the slope of its regret curve did not decrease as fast as other models’ over the course of interactions. After a careful investigation, we found this is caused by its scheduled update: because less and less updates can be performed in the later rounds, FALCON+ still selected the suboptimal arms quite often. Another important observation is that FALCON+ has a very high variance in its performance: in some epochs, it could keep sampling the suboptimal arms. This is again caused by its lazy update: in some epochs of some trials, its estimation could be biased by large reward noise, which will directly cost it miss the right arm in almost the entire next epoch. Moreover, as we discussed before, FALCON+ has a polynomial dependence on the number of arms, while NPR only has a logarithmic dependence on it. This theoretical advantage of NPR is also verified in our current results.
> > >
> > > Another issue with FALCON+’s hyper-parameter tuning is that because in practice we can only use a small portion of initial rounds to find the best set of hyper-parameters (e.g., first 1k rounds in our experiments), the selected hyper-parameters might not be consistently the best over time, given FALCON+ is epoch-based. More specifically, the model’s behavior in the earlier epochs might be indicative about its performance in the later rounds, especially in a neural network model. This adds another layer of complexity of applying FALCON+ in practice.
> > >
> > > **Is \\(\nu\\) a hyper-parameter in practice that is tuned?**
> > >
> > > As we mentioned in the response 2 (2), because \\(\nu\\) depends on the hyper-parameter \\(\delta\\), in our experiments when all the \\(K\\) arms are presented to the algorithm at once, we tuned the hyper-parameter \\(\delta\\) so that the corresponding exploration parameter \\(\nu\\) is equivalently searched. But if we only reveal a subset of the \\(K\\) arms to the algorithm each time, it is impossible to analytically calculate \\(\nu\\), because \\(\lambda_{min}\\) becomes unknown. Under the setting with a subset of arms in each round, e.g., \\(k=20\\), we tried both tuning \\(\lambda_{min}\\) and \\(\delta\\), or directly tuning \\(\nu\\), which led to similar results. In the paper, when \\(k=20\\), we reported results obtained by directly tuning \\(\nu\\).

---

> ### Author Response · Authors · 2021-11-17
> **Response to Reviewer NMyc**
>
> Due to the character limit, we response cons 5&6 in the following.
>
> 5. (1) In our synthetic setting, the purpose is to compare both the computational efficiency and effectiveness of NPR with the baselines, because our neural bandit baselines cannot afford a large NN model due to their huge time complexity. Therefore, for the neural models, we chose a relatively small network width, e.g., m = 64, to make sure that NeuralUCB/NeuralTS can be run with the full covariance matrix (described in the original paper) in our original experiments. The results showed that all the neural models achieved better performance than their linear counterparts. However, such a small neural network unfortunately failed to capture the non-linear relationship, as shown in our previous results (and pointed out by the reviewer). Now we re-run the experiments with a more complicated neural network, e.g., setting neural network width m = 128 for each hidden layer. But due to the limited GPU memory, it is now impossible to run NeuralUCB and NeuralTS with a full covariance matrix. We compared NPR and the other baselines in the newly added result figures (in Section A.1.1. of the appendix in our updated paper), and we can observe 1) all neural bandit models obtained sublinear regret; 2) NPR performed better than NeuralUCB and NeuralTS, as it does not need to employ any approximation.
> (2) To ensure visibility of our result figures, we decided not to include variances. We include the variance of the newly updated result, in Figure 4 in the appendix. We can observe the shaded areas for variance seriously overlap with each other and affect the readability of the results. Hence, we still prefer not to include them in our figures.
>
> 6. Thanks for the suggestions! As described in our response to the first question, FALCON+’s performance heavily depends on the hyper-parameter setting of its learning rate \\(\gamma_m\\). And because the tuning of \\(\gamma_m\\) has to be specific to each epoch, the current performance of FALCON+ is very poor (only slightly better than a random policy). We will try our best to improve its performance and report the results as soon as possible.

---

### Official Review · Reviewer_8GZW · 2021-11-02

**Correctness:** 4
**Technical Novelty And Significance:** 2
**Empirical Novelty And Significance:** 2
**Recommendation:** 5
**Confidence:** 4

**Main Review:**

The idea of using the reward perturbation to avoid the exploration of the entire neural network is sharp. This allows their algorithm to significantly improve over NeuralUCB and NeuralTS in practice.

However, using the reward perturbation has been exploited in previous bandit models as in generalized linear bandits. This is also mentioned by the authors in the related works. Applying this idea to the neural bandits is straightforward. In an over-parameterization regime, the neural network behaves as the linear model. So I do not see technical challenges here because the main challenges of the neural bandits have been solved in both NeuralUCB and NeuralTS.  The used techniques of this paper is an adaptation of  NeuralUCB, NeuralTS to the situation with perturbed rewards. Indeed, Lemma 4.1 is similar to Lemma 5.1 in the NeuralUCB paper. Events $E_{t,1}$, $E_{t,2}$ are similar to the events $\mathcal E^{\mu}$ and $\mathcal E^{\sigma}$ defined in the neuralTS paper. Lemma 4.2 and Lemma 4.3 are similar to Lemma 4.3 and Lemma 4.2 in the NeuralTS paper, etc.

If the authors can show me the novelty in technical proofs compared to previous works, I will raise my score.




**Summary Of The Paper:**

This paper addresses the reduction of computational cost of the neural contextual bandit problem which uses the deep neural networks to model the reward function. Existing works in this field often require the exploration of the entire neural network parameter space
and thus, the computation cost is expensive, especially when the network is large. Their main contribution is to perturb the rewards when updating the network to eliminate the need for exploration. They showed that this idea works well in experiments while still maintaining the same regret bound rate as that of previous works in the NTK regime.

**Summary Of The Review:**

This is a good paper. It brings a solution that is much more practical than NeuralUCB and NeuralTS.  The experimental results are convincing. However, I think that the novelty is not enough. I am leaning forward to rejecting it.

---

> ### Author Response · Authors · 2021-11-17
> **Response to Reviewer 8GZW**
>
> We sincerely thank the reviewer for the comments and suggestions. Please find our response in the following.
>
> We agree that our analysis on the quantification of neural network approximation error shares a similar structure with that in NeuralUCB and NeuralTS, because NPR is **based on the same assumption about the neural network**, e.g., under the neural tangent kernel regime (Lemma 4.1).
>
> Distinct from algorithm procedures in NeuralUCB and NeuralTS, NPR does not introduce anything beyond the originally required neural network update, besides perturbing the observed reward history. And hence our key technical contribution is to theoretically derive the minimum amount of added noise to help “correct” any underestimation caused by both the observation noise and the approximation error. One of the key steps is to **isolate the deviations caused by different sources in the estimated reward, e.g., the observation noise, the added pseudo noise and the neural network approximation**.
>
> To solve it, our analysis introduced the idea of **auxiliary least square solutions \\(A_t^{-1}\bar b_t\\) and \\(A_t^{-1}b_t\\)**, and define the **event \\(E_{t, 1}\\)** for the analysis of **deviation caused by the observation noise**, and **event \\(E_{t, 2}\\)** for the analysis of the **deviation caused by the neural network approximation and the added noise**. In addition, Lemma 4.2 and Lemma 4.3 are introduced to analyze concentration behavior of the deviations. All those events are **virtual** in NPR, as the algorithm itself does not need to explicitly depend on or compute any of such events.
>
> In NeuralTS, the estimated mean reward only contains **deviation caused by the neural network approximation and the observation noise**, which is analyzed by its **event \\(E_t^{\mu}\\)** and Lemma 4.3 in the NeuralTS paper. As NeuralTS does **sampling**, the corresponding deviation is analyzed by **event \\(E_t^{\sigma}\\)** and Lemma 4.2 in the NeuralTS paper. As this sampling step is independent from its neural network training, such deviation is straightforward. But the deviation caused by the added noise in NPR is generated during the neural network training, which requires more complex and careful analysis on the concentration and anti-concentration behaviors.

---

### Official Review · Reviewer_ZmVR · 2021-11-07

**Correctness:** 4
**Technical Novelty And Significance:** 3
**Empirical Novelty And Significance:** 2
**Recommendation:** 6
**Confidence:** 3

**Main Review:**

Strengths:
* Unlike the algorithms in existing literatures which are either computationally intractable or has no theoretical regret upper bound using approximate computations, NPR is computationally tractable and has nearly optimal regret upper bound.
* Extensive experiments are conducted to compare the practical performance between the proposed algorithm and the ones in existing literatures.

Weaknesses
* NPR is technically sound but hard to be applied to practical applications due to that 1) During each time step, the algorithm has to use all of the previous rewards. When the time horizon T is large, the computation becomes expensive. 2) The model studied is a K-armed setting where during each time step, the learner are facing the *same* set of arms. For practical applications, it is more reasonable to assume the arms set faced during each time step is different.
* The originality of the techniques used to prove the regret upper bound is limited. If I read the paper correctly, it seems the key lemma (Lemma 4.1) is from Jacot et al., 2018. It might be better if the authors could spend one paragraph to discuss the technique originality.
* The experiment setup seems to be different from the one discussed in this paper.

**Summary Of The Paper:**

This paper investigates neural contextual bandit problem. A new, computationally tractable and efficient algorithm (Neural bandit with perturbed reward, NPR) is proposed and proved to suffer a nearly optimal minimax regret upper bound.

**Summary Of The Review:**

Personally, I like the idea using perturbed rewards to implicitly let the neural network to explore. However the proposed algorithm is still hard to be applied to practical applications.

Detailed comments:

Equation 3.2: parameter $\lambda$ is missing?

After Lemma 4.3: ... the the neural network ...  the extra the should be removed

Lemma 4.4: it might be better if the authors could define $E_{t, 3}$ in the beginning of Lemma 4.4.

After Lemma 4.6: please resolve the cross reference error

Experiment 5.1: It is confused to me that `At each round t, only a subset of k arms out of the total K arms are sampled without replacement and discosed to all algorithms for selection.` Could the authors help explain why the algorithms are restricted to pull $k < K$ arms which seems to be different from the one discussed in this paper?

---

> ### Author Response · Authors · 2021-11-17
> **Response to Reviewer ZmVR**
>
>
> We sincerely thank the reviewer for the comments and suggestions. Please find our responses in the following.
>
> 1. **"During each time step, the algorithm has to use all of the previous rewards. When the time horizon T is large, the computation becomes expensive. "**
>
> Retraining the NN model with the complete history is required by the needed convergence of parameter estimation, which unfortunately is necessary in almost all existing neural bandit algorithms with theoretical guarantees. Therefore, NPR does not have added computational complexity on this aspect.
> It is also known that incremental model update could lead to catastrophic forgetting in the continual learning setting, of which the corresponding theoretical convergence analysis is still lacking currently. On the other hand, to enable efficient model learning, scheduled model update can be applied for the retraining process of NN (e.g., Abbasi-Yadkori, Y., Pál, D., & Szepesvári, C. (2011)). However, this is at the price of increasing regret. Finding a more efficient (online) optimizer is an important next step to further improve NPR.
>
>
> 2. **"The originality of the techniques used to prove the regret upper bound is limited."**
>
> The neural tangent kernel technique  (Lemma 4.1) is the basis of our analysis. But our key contribution is on top of it to 1) first prove perturbed rewards can sufficiently incentivize exploration in a neural network, and 2) precisely analyze the minimally required amount of noise that leads to no regret learning with a neural network model.
>
> Without our carefully analyzed noise scale, the pseudo noise with a too large variance slows down the convergence of model learning (i.e., increased regret); or the pseudo noise with a too small variance will lead to insufficient exploration and thus linear regret. Our analysis of the randomness from our added pseudo noise and the prediction error from the neural network in Lemma 4.3 (concentration) and Lemma 4.4 (anti-concentration) make the accurate calculation of the necessary amount of pseudo noise possible. Only with such level of detailed derivation and model setup one can obtain the desired sublinear regret and performance guarantee in NPR.
>
>
> 3. **"The experiment setup seems to be different from the one discussed in this paper."**
>
> We adopt the K-armed setting for the theoretical analysis, because to achieve the optimism, the noise level is required to be set at the beginning of the interactions. Or more specifically, with the K-arm setting, we can get the theoretically optimal noise level for NPR: Based on the minimum eigenvalue of the covariance matrix after K rounds (Lemma 4.4), which leads to our regret analysis.
>
> In our experiment, we also tested the model by only revealing a subset of the K arms at each round, which is more general in practice (as the reviewer suggested) and also introduces more practical difficulties. And our results show that under such a practical and more challenging setting, our model can still provide comparable performance with the state-of-the-art algorithms (by a slightly increased level of pseudo noise than the theoretical value). To provide more detailed comparison on this aspect, we also added one experiment comparing the algorithms with different values of k in the appendix. The results show that disclosing the whole arm pool to the algorithm can indeed reduce the regret of all algorithms, while NPR also works under the setting when only a subset of arms are revealed in each round. The details can be found in Section A.1.2 of the revised paper.

---

> > ### Comment · Reviewer_ZmVR · 2021-11-23
> > **Post author response**
> >
> > Thanks for the detailed response and conducting additional experiments, which solve part of my concerns. However, the proposed algorithm is still hard to be applied to practical applications. I will slightly raise my score to weakly accept. Given that there is no rating 7, I will keep my current score.

---

### Author Response · Authors · 2021-11-17
**Overall response**

We thank all reviewers for their insightful comments and suggestions! We have revised the paper to fix the typos and added new experiments in the appendix to provide more details on our original experiment settings and comparisons against the baselines.

We will respond to each reviewer’s comments individually in the following.

---

### Decision · Program_Chairs · 2022-01-20

**Decision:**

Accept (Poster)

**Comment:**

This paper studies the design and analysis of contextual bandits algorithms, combining the ideas of neural network models (Zhou et al, 2020 and Zhang et al, 2020) and reward perturbations (Kveton et al, 2019, 2020); this has the computational advantage of avoiding inverting large covariance matrices, as is done in the other neural contextual bandits algorithms. Although the reviewers think that the papers need to do a better job in highlighting differences and extra challenges in the current work compared to prior works, they also acknowledge that this paper is the first that combines the above two ideas.

The reviewers also acknowledge that the additional experiments in the rebuttal period help clear the concern the reviewers have about why all regret curves look linear. However they also pointed out, that comparison with the FALCON+ algorithm (Foster et al, 2020) may be slightly unfair, as the algorithm retrains the neural network after every new iteration. Overall, the reviewers think that the pros outweight the cons, and they lean towards acceptance.